# Speeding up fairness reductions

**Andrea Baraldi**                                    *andrea.baraldi96@unimore.it*
*University of Modena and Reggio Emilia*

**Matteo Brucato**                                         *matteo@osm-data.com*
*OSM Data*

**Miroslav Dudík**                                      *mdudik@microsoft.com*
*Microsoft*

**Francesco Guerra**                               *francesco.guerra@unimore.it*
*University of Modena and Reggio Emilia*

**Matteo Interlandi**                                   *mainterl@microsoft.com*
*Microsoft*

**Reviewed on OpenReview:** *https://openreview.net/forum?id=C0AdL3r1Dc*

## Abstract

We study the problem of fair classification, where the goal is to optimize classification accuracy subject to fairness constraints. This type of problem occurs in many real-world applications, where we seek to assure that a deployed AI system does not disproportionally impact historically disadvantaged groups. One of the leading approaches in the literature is the *reduction approach* (Agarwal et al., 2018; 2019), which enjoys many favorable properties. For instance, it supports a wide range of fairness constraints and model families and is usually easy to incorporate in existing ML pipelines. The reduction approach acts as a wrapper around a standard ML algorithm and obtains a model that satisfies fairness constraints by repeatedly running a fairness-unaware base algorithm. A typical number of iterations is around 100, meaning that the reduction approach can be up to 100 times slower than the base algorithm, which limits its applicability. To overcome this limitation, we introduce two algorithmic innovations. First, we interleave the exponentiated gradient updates of the standard reduction approach with *column-generation updates*, which leads to a decrease in the number of calls to the base algorithm. Second, we introduce *adaptive sampling*, which decreases the sizes of the datasets used in the calls to the base algorithm. We conduct comprehensive experiments to evaluate efficacy of our improvements, showing that our two innovations speed up the reduction approach by an order of magnitude without sacrificing the quality of the resulting solutions.

## 1 Introduction

As artificial intelligence (AI) systems are deployed in a growing range of applications, there is an increased need to ensure that their deployment does not disproportionately impact historically disadvantaged populations and groups (Crawford, 2013; O'Neil, 2016; Broussard, 2018; Noble, 2018; Benjamin, 2019). Fairness of AI systems is a topic of multiple academic venues,[1] a priority for regulators (The European Parliament & The Council of the European Union, 2024) as well as corporations (Crampton, 2022; Philomin, 2024; Google, 2025; Microsoft, 2025), and a focus of several open source projects (Lee & Singh, 2021).

---

[1]`https://facctconference.org/` (FAccT), `https://www.aies-conference.com/` (AIES), `https://responsiblecomputing.org/` (FORC)

There are many different kinds of fairness harms (Barocas et al., 2017; Crawford, 2017; Wallach & Dudík, 2021; Shelby et al., 2023), and many different ways to mitigate them by intervening at different points of the AI lifecycle (Wallach & Dudík, 2021), including during task definition, data collection, model training, and after model deployment. Here we study algorithmic techniques that seek to mitigate two broad categories of harms, *allocative harms* and *quality-of-service harms*, at the model training stage. Allocative harms occur when AI systems are used to allocate opportunities or resources in ways that can have significant negative impacts on people's lives such as in hiring, policing, education, and access to health-care (Angwin et al., 2016; Obermeyer et al., 2019; Raghavan et al., 2020; Smith, 2020). Quality-of-service harms occur when a system does not work as well for members of one group as it does for members of another group (Buolamwini & Gebru, 2018; Koenecke et al., 2020).

For example, suppose a hospital is training an AI model to predict 30-day hospital readmission to prioritize high-risk patients for more intensive post-discharge care. Allocative harms might occur if certain subgroups of patients (e.g., Black patients) are disproportionately under-prioritized for more intensive care (i.e., have low selection rates) or are under-selected relative to their observed rate of readmission (i.e., have high false negative rates); this type of harm has been noted, for example, by Obermeyer et al. (2019). To mitigate this harm, the hospital could train a model that incorporates suitable fairness constraints by constraining, for example, the difference between selection rates across different race groups.

Although there is a large body of research on algorithmic mitigation of fairness harms (Barocas et al., 2019; Pessach & Shmueli, 2023; Caton & Haas, 2023; Mehrabi et al., 2021), there are many frictions in applying these techniques in real world (Holstein et al., 2019), including lack of flexibility in the choice of fairness metrics and model families, lack of compatibility with existing machine learning pipelines, and computational cost of training.

In this paper, we focus on the reduction approach to unfairness mitigation (Agarwal et al., 2018; 2019), which helps address many of these usability frictions. It works with a wide range of fairness definitions, model families, and as a reduction approach, it can be "wrapped" around any existing supervised machine learning approach and so offers the flexibility to improve fairness of existing AI systems without a need to re-architect deployed systems. Because of its flexibility, it has been incorporated in several open-source toolkits, including *fairlearn* (Weerts et al., 2023) and *AIF360* (Bellamy et al., 2019). The main barrier for its broader adoption is that the training algorithm is substantially slower than fairness-unaware algorithms. Our work seeks to overcome this limitation.

We introduce two algorithmic innovations that speed up the reduction approach by several orders of magnitude. For simplicity we focus on the binary classification task and allocational harms and extend the algorithmic approach of Agarwal et al. (2018), but our innovations are also applicable in regression setting with quality-of-service harms and can be used to extend the corresponding approach of Agarwal et al. (2019), which is structurally similar to the one studied here.

The algorithm at the core of the reduction approach is the *exponentiated gradient* of Kivinen & Warmuth (1997), which we refer to as ExpGrad. The goal of ExpGrad is to find a classifier from some family (like linear classifiers or neural nets), which maximizes classification accuracy subject to fairness constraints. ExpGrad operates as a wrapper around any standard classification algorithm for the given family, which we refer to as an *oracle* or a *base algorithm*. To optimize accuracy subject to fairness constraints, ExpGrad repeatedly invokes the oracle on reweighted versions of the training data (much like boosting algorithms, Freund & Schapire, 1997), requiring up to 100 runs of the base algorithm. Thus, its running time is around 100 times slower than the running time of fairness-unaware methods.

Our first innovation is in decreasing the number of iterations of the optimization procedure and hence the number of oracle calls to the base algorithm with the use of *column generation*. Column generation (Eisemann, 1957; Griva et al., 2008) is a classical optimization approach particularly suited for solving large linear programming problems with a special structure like the one studied here. While column generation tends to work well in practice, we are not aware of any convergence guarantees similar to those enjoyed by ExpGrad. We show how to combine the two approaches to obtain the best of both worlds. In practice, column generation decreases the number of ExpGrad iterations from around 100 to around 10 (see our experiments in Section 5) and the combined approach retains the convergence guarantees of ExpGrad.

While the column generation decreases the number of oracle calls, the goal of our second innovation is to decrease the cost of each oracle call. One simple approach, which we call *static sampling*, would be to subsample the training data uniformly at random and solve the constrained optimization problem for the smaller dataset; this improves the running time but leads to some loss in accuracy. We improve upon this naïve strategy by noting that datasets passed to the oracle are weighted, with a different weighting in each iteration. By sampling the data adaptively, according to the weights, we sacrifice less accuracy than we would if we used static sampling. For instance, if the dataset weights generated in a given iteration of ExpGrad put 90% of probability mass on 10% of examples, then picking the 10% examples with the largest weights results in a much smaller loss in accuracy than picking an arbitrary 10% of examples. In our experiments, we show that our adaptive sampling approach outperforms static sampling, and we do not see major losses in accuracy, even when the sample size is as small as 10% of the original training set (see Section 5).

Our experiments evaluate the efficacy of ExpGrad$^{++}$ and compare it both with ExpGrad and other baselines. We show that our two innovations (column generation and adaptive sampling) speed up ExpGrad by an order of magnitude. The resulting approach still enjoys the flexibility of ExpGrad, while substantially improving its scalability.

## 1.1 Usage guidelines, risks, and limitations

This paper studies a reduction-based approach to unfairness mitigation. Before using it in practice, it is essential to consider the societal context of the application (Green, 2021; Selbst et al., 2019), and pay attention to the entire AI lifecycle not just the model training stage, since choices in other stages (like data collection and task definition) could outweight any benefits from unfairness mitigation in model training. In some contexts, the best unfairness mitigation might be to avoid a technological intervention altogether (Baumer & Silberman, 2011).

The reduction-based approach seeks to optimize a tradeoff between fairness and accuracy. The tradeoff-based framing could be problematic for various reasons (Cooper et al., 2021), including overreliance on the mathematical formalization of fairness and accuracy. To help mitigate the risk of overreliance, practitioners should not blindly deploy the model returned by an algorithm. Instead, multiple models along the fairness-accuracy frontier should be evaluated using relevant metrics on relevant subpopulations (Barocas et al., 2021). Any substantial tradeoffs along the frontier should be analyzed. They might point to data issues requiring non-algorithmic interventions, such as gathering of additional (less biased) data or introduction of new features (Chen et al., 2018).

In high-stakes applications it is crucial to directly examine the classification rules before they are deployed. The reduction-based approach returns an ensemble of base models, which is harder to interpret than individual base models. Moreover, the predictions outputted by ensembles are randomized, which can be problematic in some applications. To deal with both of these issues, we suggest that practitioners should consider each individual model in the ensemble separately, and among all of them choose the most suitable one, based both on evaluation of relevant metrics as well as on direct examination of the model.

The two speed-up strategies introduced in this paper are applicable to the reduction-based approach to fair classification (Agarwal et al., 2018), and also to the reduction-based approach to fair regression, when mitigating quality-of-service harms (Section 5 of Agarwal et al., 2019). The empirical evaluation in this paper does not cover all of these settings. Our experiments only cover binary classification and two fairness metrics (demographic parity and equalized odds). Applications to other settings should be validated by empirically comparing with appropriate baselines.

Similarly, although the reduction-based approach works with a wide range of base learners and data types, our empirical evaluation focuses on tabular data and two base learners (logistic regression and boosted decision trees), so applications to other data types and learners should be validated empirically as well.

Finally, while column generation can dramatically decrease the number of oracle calls to the base learner, in our experiments it still requires 10–20 calls, which can be prohibitively expensive when training very large models.

## 2    Related work

We conceptualize fairness through the lens of fairness harms (Crawford, 2017; Wallach & Dudík, 2021; Shelby et al., 2023), by which we mean negative impacts on groups of people, such as those defined in terms of race, gender, age, or disability status. In fairness literature, this is also referred to as group fairness (Dwork et al., 2012). There are various alternative frameworks of fairness of AI systems, which we do not pursue here, such as individual fairness (Dwork et al., 2012) and fairness based on causal reasoning (Loftus et al., 2018).

Many algorithmic mitigation approaches have been proposed in the literature (see, e.g., the surveys of Mehrabi et al., 2021; Caton & Haas, 2023; Pessach & Shmueli, 2023). These can be broadly divided into three categories according to when they are applied relative to model training (Mehrabi et al., 2021; Caton & Haas, 2023; Barocas et al., 2019; Islam et al., 2022a; Pessach & Shmueli, 2023). *Pre-processing* techniques seek to improve fairness by performing changes in the training data before passing it to an ML algorithm (Calmon et al., 2017; Feldman et al., 2015; Kamiran & Calders, 2012). *In-training* (or *in-processing*) techniques modify the ML algorithms to account for fairness during the training time (Zafar et al., 2017a;b; Woodworth et al., 2017; Kamishima et al., 2011; Zhang et al., 2018; Cruz et al., 2023). *Post-processing* techniques analyze and modify the outputs of an already trained model (Hardt et al., 2016; Cruz & Hardt, 2024).

Pre-processing techniques are agnostic to the choice of the model family and typically also to the choice of fairness constraints. In practice this may lead to both suboptimal accuracy and suboptimal fairness (see our experiments as well as those of Agarwal et al., 2018). Post-processing techniques are designed to optimize tradeoff between fairness and accuracy and are also flexible in the choice of the model family, but they require access to the sensitive attribute at the deployment time. This may be inappropriate to use, or in some domains may be prohibited by law.[2] In-training techniques are capable of achieving optimal fairness–accuracy tradeoffs and do not require access to the sensitive attribute at the deployment time. However, many in-training techniques only incorporate one specific type of fairness constraint in one specific training algorithm (and one specific model family). This limits their usability in real-world scenarios where assessing the dataset's fairness using various metrics or testing different machine learning models is necessary. There are some exceptions to this. One of them is the reduction approach (Agarwal et al., 2018; 2019), which we study here. Another exception are direct optimization approaches, which assume the ability to model certain conditional densities (Menon & Williamson, 2018; Celis et al., 2019) and based on these directly optimize the accuracy subject to constraints. However, modeling such conditional densities is statistically feasible only when the data dimensionality is small, limiting the applicability of these techniques.

Another recent family of techniques, which broadly falls in the in-training category, are automated machine learning (AutoML) approaches that search over a space of models and hyperparameters to optimize both accuracy and fairness metrics (see Weerts et al., 2024, for an extensive discussion of work in this subfield, including usage guidelines, opportunities, and risks). Similar to reductions, AutoML approaches work with generic model families and can be easily included in existing pipelines, and additionally they offer greater flexibility when it comes to the choice of fairness metrics. However, generality of their optimization algorithms comes at a cost: their convergence guarantees are typically much weaker (for instance, with local rather than global convergence guarantees and worse dependence on the dimension). This can be mitigated by incorporating reductions as one of the algorithms considered within an AutoML framework, potentially getting the "best of all worlds" (from the optimization perspective).

## 3    Preliminaries

### 3.1    Problem definition

We consider binary classification tasks, where the input is a dataset consisting of labeled examples $(X_1, A_1, Y_1), \ldots, (X_n, A_n, Y_n)$, where $X_i \in \mathbb{R}^d$ is a feature vector describing the $i$-th example, $A_i \in \mathcal{A}$ is a categorical sensitive attribute, and $Y_i \in \{0, 1\}$ is a binary label.

---

[2]For example, under the U.S. Equal Credit Opportunity Act (15 U.S.C. 1691 et seq.), which regulates lending, it is possible to use the applicant's age (in a narrow sense described in Regulation B, 12 C.F.R. part 1026), but the use of other sensitive attributes like race or gender is prohibited.

Consider the example from Section 1, in which the goal is to train an AI system that refers patients into a post-discharge care program based on the risk of a 30-day readmission. In this case, $X_i$ contains clinical information about a patient; $A_i$ is a categorical variable encoding patient's race and ethnicity; and $Y_i$ indicates whether the patient was readmitted within 30 days of discharge.

The goal of a classification algorithm is to produce a classifier $h : \mathbb{R}^d \to \{0, 1\}$ that accurately predicts label $Y$ on a new example represented by a feature vector $X \in \mathbb{R}^d$. We assume that $h$ is chosen from some family $\mathcal{H}$ (like linear classifiers, decision trees, or neural nets). For example, logistic regression considers linear classifiers of the form $h_\beta(X) = 1\{\beta^T X \geq 0\}$ where $\beta \in \mathbb{R}^d$ and $1\{\cdot\}$ is an indicator function.

The sensitive attribute $A$ is only required during training, but not during inference. However, since we make no assumptions about the relationship between $X$ and $A$, it is possible for $X$ to contain some information about $A$. In this way, our task definition encompasses both the settings when the sensitive attribute is available at inference time (in that case it is included as part of $X$) as well as the settings when the sensitive attribute is not available (in that case it is not included as part of $X$).

Standard classification algorithms seek $h \in \mathcal{H}$ that minimizes training classification error $err(h)$:[3]

$$\min_{h \in \mathcal{H}} err(h), \quad \text{where} \quad err(h) = \frac{1}{n} \sum_{i=1}^{n} 1\{h(X_i) \neq Y_i\}. \tag{1}$$

Algorithmic approaches to unfairness mitigation instead optimize training classification error under a fairness constraint, typically specified using the sensitive attribute $A$.

There are many notions of (un)fairness, appropriate in different applications (for instance, Islam et al., 2022a, list 34 fairness notions collected from a review of the literature). In this paper, for simplicity, we focus on two standard quantitative definitions of fairness, but our technique encompasses many other notions (see Agarwal et al., 2018, for further details). Specifically, we consider *demographic parity* and *equalized odds* (Hardt et al., 2016; Barocas et al., 2019):

**Definition 1** (Demographic parity: DP)**.** We say that a classifier $h$ satisfies *demographic parity* with respect to a distribution over triples $(X, A, Y)$ if its decision is statistically independent of $A$, that is, if $\mathbb{E}[h(X) \,|\, A = a] = \mathbb{E}[h(X)]$ for all $a \in \mathcal{A}$.

**Definition 2** (Equalized odds: EO)**.** We say that a classifier $h$ satisfies *equalized odds* with respect to a distribution over triples $(X, A, Y)$ if its decision is statistically independent of $A$, conditional on $Y$, that is, if $\mathbb{E}[h(X) \,|\, A = a, Y = y] = \mathbb{E}[h(X) \,|\, Y = y]$ for all $a \in \mathcal{A}$ and $y \in \{0, 1\}$.

The degree to which a classifier $h$ satisfies demographic parity on a data set $(X_1, A_1, Y_1), \ldots, (X_n, A_n, Y_n)$ can be quantified using the quantity

$$\Delta_{\mathrm{DP}}(h) = \max_{a \in \mathcal{A}} \left| \hat{\mathbb{E}}[h(X) \,|\, A = a] - \hat{\mathbb{E}}[h(X)] \right| \tag{2}$$

$$= \max_{a \in \mathcal{A}} \left| \frac{1}{n_a} \sum_{i:\, A_i = a} h(X_i) - \frac{1}{n} \sum_{i=1}^{n} h(X_i) \right|,$$

where $n_a$ is the number of examples with $A_i = a$. Compared with Definition 1, in Eq. (2) we replace true expectations $\mathbb{E}[\cdot]$ with empirical averages $\hat{\mathbb{E}}[\cdot]$, evaluated on the training data. The value of $\Delta_{\mathrm{DP}}$ is equal to the largest deviation of $\hat{\mathbb{E}}[h(X) \,|\, A = a]$ from $\hat{\mathbb{E}}[h(X)]$, across all $a$. Definition 1 is satisfied (on the empirical distribution) exactly when the deviations across all $a$ are equal to zero. Otherwise, $\Delta_{\mathrm{DP}}$ quantifies an (additive) violation of fairness constraints. We refer to $\Delta_{\mathrm{DP}}$ as the *DP difference.*

---

[3]Standard algorithms also include various mechanisms to control overfitting, such as regularization or early stopping. Similarly to Agarwal et al. (2018), we model regularization (and other similar mechanisms) by assuming that the family $\mathcal{H}$ has been appropriately restricted. For example, for regularized logistic regression, $\mathcal{H} = \{h_\beta : \beta \in \mathbb{R}^d, \|\beta\| \leq C\}$, where the value of $C$ controls overfitting.

For equalized odds, we can quantify the violation of fairness constraints using an analogous quantity, called the *EO difference*:

$$\Delta_{\mathrm{EO}}(h) = \max_{a \in \mathcal{A}, y \in \{0,1\}} \left| \hat{\mathbb{E}}[h(X) \mid A = a, Y = y] - \hat{\mathbb{E}}[h(X) \mid Y = y] \right| \tag{3}$$

$$= \max_{a \in \mathcal{A}, y \in \{0,1\}} \left| \frac{1}{n_{a,y}} \sum_{i:\, A_i = a, Y_i = y} h(X_i) - \frac{1}{n_y} \sum_{i:\, Y_i = y} h(X_i) \right|,$$

where $n_y$ and $n_{a,y}$ refer to the number of examples with $Y_i = y$ and $A_i = a$, $Y_i = y$, respectively.

Continuing with the post-discharge care example, demographic parity states that the patients from all race/ethnicity groups should be referred to the post-discharge care program at equal rates. The fairness constraint $\Delta_{\mathrm{DP}}(h) \leq \epsilon$ states that the referral rate of every race/ethnicity group is only allowed to differ from the overall referral rate by at most $\epsilon$.

The equalized odds condition states that false positive rates and false negative rates for all the race/ethnicity groups should be the same. The fairness constraints $\Delta_{\mathrm{EO}}(h) \leq \epsilon$ states that false positive rates and false negative rates of every race/ethnicity group are allowed to differ from the overall false positive rates and overall false negative rates, respectively, by at most $\epsilon$.

## 3.2 Reduction approach

Reduction approach to fair classification (Agarwal et al., 2018) seeks to solve the problems of the form:

$$\min_{h \in \mathcal{H}} err(h) \quad \text{such that} \quad \Delta(h) \leq \epsilon, \tag{4}$$

where $\Delta$ formalizes fairness constraints and $\epsilon \geq 0$ is the degree to which we allow the fairness constraint to be violated.

Reduction approach works with a broad family of fairness constraints including $\Delta_{\mathrm{DP}}$, $\Delta_{\mathrm{EO}}$ and many others (see Agarwal et al., 2018, for details). It also works with any family of ML models; it only requires the ability to solve weighted (but unconstrained) classification problems, meaning problems that minimize weighted classification error

$$\min_{h \in \mathcal{H}} \frac{1}{n} \sum_{i=1}^{n} w_i \mathbf{1}\{h(X_i) \neq Y_i\} \tag{5}$$

for any set of weights $w_i \geq 0$. Eq. (5) can be solved by standard fairness-unaware classification algorithms, like those that fit logistic regression models, decision trees, or neural nets. The algorithms that solve Eq. (5) are viewed as *oracles* from the perspective of the reduction, and in practice they are implemented by library calls.

In order to leverage the strength of an oracle, we cast the constrained optimization problem in Eq. (4) as a linear optimization. For a classifier $h \in \mathcal{H}$, let $\mathbf{v}_h \in \mathbb{R}^n$ denote the vector with entries $v_{h,i} = h(X_i)$ for $i = 1, \ldots, n$. Then Eq. (4) can be rewritten as

$$\min_{h \in \mathcal{H}} \mathbf{c}^T \mathbf{v}_h + c_0 \quad \text{such that} \quad \mathbf{A}\mathbf{v}_h \leq \mathbf{b}, \tag{6}$$

where the vector $\mathbf{c} \in \mathbb{R}^n$ and the scalar $c_0 \in \mathbb{R}$ are determined by the choice of an error metric, in our case $err(h)$; the matrix $\mathbf{A} \in \mathbb{R}^{k \times n}$ and vector $\mathbf{b} \in \mathbb{R}^k$ are based on the choice of the fairness metric $\Delta$ and bound $\epsilon$ from Eq. (4). The dimension $k$ is determined by the cardinality of $\mathcal{A}$ and the choice of the fairness metric $\Delta$.

For example, using Eq. (2), the constraint $\Delta_{\mathrm{DP}}(h) \leq \epsilon$ can be written as $2|\mathcal{A}|$ constraints of the form

$$\hat{\mathbb{E}}[h(X) \mid A = a] - \hat{\mathbb{E}}[h(X)] \leq \epsilon$$

$$-\hat{\mathbb{E}}[h(X) \mid A = a] + \hat{\mathbb{E}}[h(X)] \leq \epsilon$$

for all $a \in \mathcal{A}$, which can be expressed using a suitable matrix $\mathbf{A}$, with the vector $\mathbf{b}$ having all entries equal to $\epsilon$. The constraint $\Delta_{\mathrm{EO}}(h) \leq \epsilon$ can be similarly expressed using $k = 4|\mathcal{A}|$ constraints (two constraints for

---

**Algorithm 1** ExpGrad

---

Input: Lagrangian specified by $\mathbf{c} \in \mathbb{R}^n$, $\mathbf{A} \in \mathbb{R}^{k \times n}$, $\mathbf{b} \in \mathbb{R}^k$;
bound $B$, learning rate $\eta$, convergence threshold $\nu$, maximum iterations $T$

1: Set $\boldsymbol{\theta}_1 = \mathbf{0} \in \mathbb{R}^k$
2: **for** $t = 1, 2, \ldots, T$ **do**
3:     Set $\lambda_{t,j} = B \frac{\exp\{\theta_{t,j}\}}{1 + \sum_{j'=1}^{k} \exp\{\theta_{t,j'}\}}$ for $j = 1, \ldots, k$
4:     $h_t \leftarrow \text{Best}_h(\boldsymbol{\lambda}_t)$, and let $\mathbf{v}_t = \mathbf{v}_{h_t}$
5:     $\mathbf{v}_{\text{EG}} \leftarrow \frac{1}{t} \sum_{t'=1}^{t} \mathbf{v}_{t'}$, $\boldsymbol{\lambda}_{\text{EG}} \leftarrow \frac{1}{t} \sum_{t'=1}^{t} \boldsymbol{\lambda}_{t'}$
6:     $\nu_{\text{EG}} \leftarrow \text{EvaluateDualityGap}(\mathbf{v}_{\text{EG}}, \boldsymbol{\lambda}_{\text{EG}})$
7:     **if** $\nu_{\text{EG}} \leq \nu$ or $t = T$ **then**
8:         Return $p$ that randomizes uniformly over $h_1, \ldots, h_t$
9:     Set $\boldsymbol{\theta}_{t+1} = \boldsymbol{\theta}_t + \eta \left( \mathbf{A}\mathbf{v}_t - \mathbf{b} \right)$

10: **function** EvaluateDualityGap($\mathbf{v}, \boldsymbol{\lambda}$)
11:     $\overline{L} \leftarrow L(\mathbf{v}, \boldsymbol{\lambda}^*)$ where $\boldsymbol{\lambda}^* = \arg\max_{\boldsymbol{\lambda}' \in \mathbb{R}_+^k, \|\boldsymbol{\lambda}'\|_1 \leq B} L(\mathbf{v}, \boldsymbol{\lambda}')$
12:     $\underline{L} \leftarrow L(\mathbf{v}_{h^*}, \boldsymbol{\lambda})$ where $h^* = \text{Best}_h(\boldsymbol{\lambda})$
13:     Return $\max\{L(\mathbf{v}, \boldsymbol{\lambda}) - \underline{L}, \overline{L} - L(\mathbf{v}, \boldsymbol{\lambda})\}$

14: **function** Best$_h(\boldsymbol{\lambda})$          // *returns* $\arg\min_{h \in \mathcal{H}} L(\mathbf{v}_h, \boldsymbol{\lambda})$
15:     Let $\mathbf{w} = \mathbf{c} + \mathbf{A}^T \boldsymbol{\lambda}$
16:     Find $h^* = \arg\min_{h \in \mathcal{H}} \mathbf{w}^T \mathbf{v}_h$
        by calling a standard classification algorithm for $\mathcal{H}$
        on the data set reweighted according to $\mathbf{w}$
17:     Return $h^*$

---

each combination of $a \in \mathcal{A}$, $y \in \{0, 1\}$). (See Agarwal et al. (2018) for full derivation of $\mathbf{A}$ for DP, EO, as well as for more general fairness constraints.) We view $k$ as a problem-specific constant. In much prior work, the sensitive attribute is binary ($|\mathcal{A}| = 2$), yielding $k = 4$ for DP and $k = 8$ for EO.

Instead of working with classifiers $h \in \mathcal{H}$, reduction approach considers a larger set consisting of randomized classifiers that randomize over a finite subset of $\mathcal{H}$. We write $\mathcal{P}(\mathcal{H})$ for the set of such randomized classifiers and identify them with the corresponding probability distributions over $\mathcal{H}$. A randomized classifier $p \in \mathcal{P}(\mathcal{H})$ makes a prediction on an example $X$ by first sampling $h \sim p$ and then predicting $h(X)$. For example, if $p$ puts 0.5 probability mass on $h_1$ and 0.5 probability mass on $h_2$, then the randomized classifier specified by $p$ predicts 1 on points $X$ where $h_1(X) = h_2(X) = 1$, predicts 0 on points $X$ where $h_1(X) = h_2(X) = 0$, and flips a coin on points $X$ where $h_1(X) \neq h_2(X)$.

The linear optimization from Eq. (6) can be generalized to randomized classifiers. For any $p \in \mathcal{P}(\mathcal{H})$, we set $\mathbf{v}_p = \sum_{h \in \mathcal{H}} p(h)\mathbf{v}_h$, and the resulting optimization problem is

$$\min_{p \in \mathcal{P}(\mathcal{H})} \mathbf{c}^T \mathbf{v}_p \quad \text{such that} \quad \mathbf{A}\mathbf{v}_p - \mathbf{b} \leq 0. \tag{7}$$

This constrained optimization problem can be algorithmically solved by finding a solution to the min-max problem

$$\min_{p \in \mathcal{P}(\mathcal{H})} \max_{\boldsymbol{\lambda} \in \mathbb{R}_+^k, \|\boldsymbol{\lambda}\|_1 \leq B} L(\mathbf{v}_p, \boldsymbol{\lambda}) \tag{8}$$

where $L(\mathbf{v}, \boldsymbol{\lambda})$ is the Lagrangian

$$L(\mathbf{v}, \boldsymbol{\lambda}) = \mathbf{c}^T \mathbf{v} + \boldsymbol{\lambda}^T (\mathbf{A}\mathbf{v} - \mathbf{b}), \tag{9}$$

and $\boldsymbol{\lambda} \in \mathbb{R}_+^k$ is the vector of non-negative Lagrange multipliers.

Conceptually, the Lagrangian *scalarizes* the original constrained optimization problem in Eq. (7) by summing up the objective and individual constraint violations, multiplied by the Lagrange multipliers. The Lagrange

multipliers $\lambda_j$, for $j = 1, \ldots, k$, specify the importance of not violating each of the constraints. The possibility of using sufficiently large $\lambda_j$ in the inner maximization forces the outer minimization to choose the point $p$ that (approximately) satisfies the constraints. It can be shown formally that with an appropriate choice of $B$, the solution of Eq. (8) approximately solves Eq. (7) (see Agarwal et al., 2018).

Reduction approach solves the min-max problem from Eq. (8) by an iterative algorithmic scheme developed by Freund & Schapire (1996) for finding an equilibrium in a zero-sum game. The min-max problem in Eq. (8) can be viewed as a game between a $\boldsymbol{\lambda}$-player seeking to maximize the Lagrangian and $\mathbf{v}$-player seeking to minimize the Lagrangian. Freund & Schapire (1996) propose an iterative protocol where in each round $t$, the $\boldsymbol{\lambda}$-player plays the action $\boldsymbol{\lambda}_t$ according to a suitable online learning algorithm (in our case the exponentiated gradient algorithm of Kivinen & Warmuth, 1997) and $\mathbf{v}$-player plays the best response $\mathbf{v}_t$ to the other player's action $\boldsymbol{\lambda}_t$. Freund & Schapire (1996) show that the averages of the played actions $\boldsymbol{\lambda}_t$ and $\mathbf{v}_t$ converge to an equilibrium of the game, which coincides with the solution of the min-max problem.

This scheme is implemented in Algorithm 1. The vector $\boldsymbol{\theta}_t \in \mathbb{R}^k$ is used to obtain $\boldsymbol{\lambda}_t$ (the action of the $\boldsymbol{\lambda}$-player) via a soft-max transformation that guarantees that the components of $\boldsymbol{\lambda}_t$ are non-negative and sum to at most $B$ (step 3). The best response of the $\mathbf{v}$-player is obtained in step 4 by calling the function $\mathrm{BEST}_h(\boldsymbol{\lambda}_t)$, which is implemented by calling the oracle for the family $\mathcal{H}$. In steps 5 and 6, we consider the current average play of the $\boldsymbol{\lambda}$- and $\mathbf{v}$-player and check how close these averages are to the equilibrium of the game by evaluating how much each player can unilaterally improve their objective (see function $\mathrm{EVALUATEDUALITYGAP}$). If the possible improvement is below a convergence threshold $\nu$, or if we have reached the maximum number of iterations, we return the current average play. Otherwise, we update $\boldsymbol{\theta}_t$, following the exponentiated-gradient update rule. Conceptually, we form a vector of constraint violations $\mathbf{A}\mathbf{v}_t - \mathbf{b}$ and increase the values of the components of $\boldsymbol{\theta}_t$ (and thus also of $\boldsymbol{\lambda}_t$) according to how much violation occurs. This means that in the next iteration of the protocol, the constraints that were more violated receive more importance in the Lagrangian.

Agarwal et al. (2018) show that for typical families $\mathcal{H}$ (like linear classifiers, neural nets, and boosted trees), given a dataset of size $n$, we should set $B \propto \sqrt{n}$, $\eta \propto 1/n$, $\nu \propto 1/\sqrt{n}$, and $T \propto n^2$ to guarantee that the solution returned by the algorithm satisfies the fairness constraints and matches the accuracy of the solution of Eq. (8) (up to an error of at most $O(1/\sqrt{n})$, which is on the same scale as the difference between the training error and error with respect to the true distribution). This means that for a dataset of size $1\,000$, the theory requires around 1 million iterations! The most costly operation in each iteration is the call to $\mathrm{BEST}_h$, which involves calling an oracle, on a weighted dataset of size $n$. So theory would yield the running time that is 1 million times slower than the running time of a fairness-unaware approach. In practice, on the datasets of size up to $50\,000$, we find that around 100 iterations suffice to reach the termination condition (see our experiments in Section 5). However, a 100-fold slowdown is still prohibitive, and presents the main obstacle for applying reduction approach with larger datasets.

## 4 Our approach: $\mathrm{EXPGRAD}^{++}$

We introduce two innovations to speed up Algorithm 1. First, we interleave exponentiated gradient with column generation (see, e.g., Griva et al., 2008, Section 7.3) to decrease the number of iterations to around 5–10 (instead of 100). Second, we use sampling to generate smaller training datasets when calling the oracle in $\mathrm{BEST}_h$. The resulting approach is presented in Algorithm 2, with new and revised steps marked with an asterisk.

### 4.1 Column generation

Assume that the family $\mathcal{H}$ is of finite cardinality, $|\mathcal{H}| = N$; this is without loss of generality, because the optimization in Eqs. (7) and (8) only considers predictions on a fixed dataset of size $n$, and hence we can assume $|\mathcal{H}| \leq 2^n$ (from the perspective of optimization).

Let $\mathbf{V} \in \mathbb{R}^{n \times N}$ be the matrix with columns corresponding to vectors $\mathbf{v}_h$ across $h \in \mathcal{H}$. Then a probability distribution $p \in \mathcal{P}(\mathcal{H})$ can be viewed as a vector in $\mathbb{R}_+^N$ with the components $p(h)$ summing to one, $\sum_{h \in \mathcal{H}} p(h) = 1$, and $\mathbf{v}_p$ is obtained by matrix-vector multiplication as $\mathbf{v}_p = \mathbf{V}p$. Thus, Eq. (8) can be

written as a linear programming (LP) problem:

$$\max_{\boldsymbol{\lambda}\in\mathbb{R}^k_+,\|\boldsymbol{\lambda}\|_1\leq B}\ \min_{p\in\mathbb{R}^N_+,\sum_{h\in\mathcal{H}}p(h)=1}\left[\mathbf{c}^T\mathbf{V}p+\boldsymbol{\lambda}^T(\mathbf{A}\mathbf{V}p-\mathbf{b})\right]. \tag{10}$$

It is intractable to solve this problem with standard LP solvers (since $N$ is exponential in $n$). To circumvent the dimensionality of $N$, the column generation (CG) approach considers a small subset of base classifiers $\tilde{\mathcal{H}}\subseteq\mathcal{H}$, with $m=|\tilde{\mathcal{H}}|$, and replaces the large matrix $\mathbf{V}\in\mathbb{R}^{n\times N}$ by a smaller matrix $\hat{\mathbf{V}}\in R^{n\times m}$ that only contains columns corresponding to the classifiers $h\in\tilde{\mathcal{H}}$. Instead of directly solving Eq. (10), CG solves a smaller restricted problem

$$\max_{\boldsymbol{\lambda}\in\mathbb{R}^k_+,\|\boldsymbol{\lambda}\|_1\leq B}\ \min_{p\in\mathbb{R}^m_+,\sum_{h\in\tilde{\mathcal{H}}}p(h)=1}\left[\mathbf{c}^T\tilde{\mathbf{V}}p+\boldsymbol{\lambda}^T(\mathbf{A}\tilde{\mathbf{V}}p-\mathbf{b})\right]. \tag{11}$$

The CG approach starts with $m=1$ and $\tilde{\mathcal{H}}$ containing an arbitrary base classifier (for example, the classifier obtained by optimizing accuracy without any fairness constraints), and then repeatedly solves the restricted problem in Eq. (11). The solution to the restricted problem is checked for optimality using the function EVALUATEDUALITYGAP from Algorithm 1. If the duality gap is non-zero, it means that $h^*$ obtained in the call to EVALUATEDUALITYGAP can be used to reduce the objective in Eq. (11), and so $h^*$ is added to $\tilde{\mathcal{H}}$, its corresponding vector $\mathbf{v}_{h^*}$ is added to $\tilde{\mathbf{V}}$, and the LP in Eq. (11) is solved again. This is repeated until convergence.

In practice CG can converge in a very small number of iterations, but we are not aware of any non-trivial theoretical upper bounds. Therefore, we interleave column generation (CG) with exponentiated gradient (EG), which allows us to benefit from strong practical performance of CG while retaining the worst-case guarantees of EG.

To combine the two approaches, we introduce a data structure that contains all of the classifiers returned by BEST$_h$ so far, which we refer to as *cache*. This *cache* plays a role of $\tilde{\mathcal{H}}$ in CG. The modified algorithm (Algorithm 2) initializes *cache* with the classifier returned by a standard (fairness-unaware) classification algorithm for $\mathcal{H}$ (step 2). In each iteration $t$, the algorithm first performs the EG update (steps 4–9), and if the convergence condition in step 8 is not satisfied, it runs a single iteration of CG with $\tilde{\mathcal{H}}=cache$ (steps 10–14). If the CG convergence condition is not satisfied (in step 13) then it finishes the EG update (in step 15) and proceeds to next iteration. Since the EG iterates are unaffected, we retain the original convergence guarantee. However, by introducing CG steps, it is possible to terminate much earlier if the duality gap of the CG iterates falls below $\nu$. In practice, we observe that this termination condition is reached in around 10 iterations instead of 100 iterations that were required by EG to reach a similar quality of the solution (see Section 5).

### 4.2 Subsampling

While CG decreases the number of iterations and therefore the number of oracle calls, the goal of our second innovation is to decrease the cost of each oracle call. We do this by subsampling the data before passing it to the oracle.

Subsampling is a general and widely used strategy that makes the training process faster and more manageable in terms of storage and memory resources, but it can lead to loss of accuracy in the obtained solution if the sample does not represent the original data sufficiently well. A simple strategy, which we call *static sampling*, is to subsample the training data uniformly at random and solve the constrained optimization problem for the smaller dataset. We expect this strategy to do well in approximating the overall training error, but it might severely impact the accuracy of fairness metrics, which are calculated on various subgroups of data, and whose sizes might become too small as a result of subsampling.

We improve upon this naive strategy by noting that datasets passed to the oracle are weighted, with a different weighting in each iteration determined by the current vector of Lagrange multipliers $\boldsymbol{\lambda}_t$ (see step 4 and the implementation of the function BEST$_h$ in Algorithm 1). The components of the weight vector $\mathbf{w}\in\mathbb{R}^n$

---

**Algorithm 2** EXPGRAD$^{++}$

---

Input: Lagrangian specified by $\mathbf{c} \in \mathbb{R}^n$, $\mathbf{A} \in \mathbb{R}^{k \times n}$, $\mathbf{b} \in \mathbb{R}^k$;
        bound $B$, learning rate $\eta$, convergence threshold $\nu$, maximum iterations $T$, sampling ratio $\rho$

  1: Set $\boldsymbol{\theta}_1 = \mathbf{0} \in \mathbb{R}^k$
*2: $cache \leftarrow \{h_{\text{init}}\}$ where $h_{\text{init}} = \arg\min_{h \in \mathcal{H}} err(h)$
  3: **for** $t = 1, 2, \ldots, T$ **do**
  4:     Set $\lambda_{t,j} = B \frac{\exp\{\theta_{t,j}\}}{1 + \sum_{j'=1}^{k} \exp\{\theta_{t,j'}\}}$ for $j = 1, \ldots, k$
  5:     $h_t \leftarrow \text{BEST}_h(\boldsymbol{\lambda}_t)$, and let $\mathbf{v}_t = \mathbf{v}_{h_t}$
  6:     $\mathbf{v}_{\text{EG}} \leftarrow \frac{1}{t} \sum_{t'=1}^{t} \mathbf{v}_{t'}$, $\boldsymbol{\lambda}_{\text{EG}} \leftarrow \frac{1}{t} \sum_{t'=1}^{t} \boldsymbol{\lambda}_{t'}$
  7:     $\nu_{\text{EG}} \leftarrow \text{EVALUATEDUALITYGAP}(\mathbf{v}_{\text{EG}}, \boldsymbol{\lambda}_{\text{EG}})$      *// see Algorithm 1*
  8:     **if** $\nu_{\text{EG}} \leq \nu$ **then**
  9:         Return $p$ that randomizes uniformly over $h_1, \ldots, h_t$
*10:     Let $\tilde{\mathbf{V}}$ be the matrix with columns $\mathbf{v}_h$ across $h \in cache$
*11:     $(\boldsymbol{\lambda}_{\text{CG}}, p_{\text{CG}}) \leftarrow$ solve Eq. (11) with $\tilde{\mathcal{H}} = cache$
*12:     $\nu_{\text{CG}} \leftarrow \text{EVALUATEDUALITYGAP}(\tilde{\mathbf{V}} p_{\text{CG}}, \boldsymbol{\lambda}_{\text{CG}})$     *// see Algorithm 1*
*13:     **if** $\nu_{\text{CG}} \leq \nu$ or $t = T$ **then**
*14:         Return $p$ that randomizes over $cache$ according to $p_{\text{CG}}$
 15:     Set $\boldsymbol{\theta}_{t+1} = \boldsymbol{\theta}_t + \eta(\mathbf{A}\mathbf{v}_t - \mathbf{b})$

 16: **function** $\text{BEST}_h(\boldsymbol{\lambda})$     *// approximates* $\arg\min_{h \in \mathcal{H}} L(\mathbf{v}_h, \boldsymbol{\lambda})$
 17:     Let $\mathbf{w} = \mathbf{c} + \mathbf{A}^T \boldsymbol{\lambda}$
*18:     Set $q_i = \frac{|w_i|}{\sum_{i=1}^{n} |w_i|}$ for $i = 1, \ldots, n$
*19:     $D \leftarrow \{\}$
*20:     **repeat** $\lceil \rho n \rceil$ times
*21:         Sample $i \sim \mathbf{q}$
*22:         Add to $D$ an example $(X, Y)$ with $X = X_i$, $Y = 1\{w_i < 0\}$
*23:     Find $h^* = \arg\min_{h \in \mathcal{H}} \sum_{(X,Y) \in D} 1\{h(X) \neq Y\}$
           by calling a standard classification algorithm for $\mathcal{H}$
*24:     Add $h^*$ to $cache$ and return $h^*$

---

specify how important each data point is. Operationally, this tends to upweight the groups for which the fairness constraint is most violated. This suggests a natural adaptive sampling strategy, which subsamples the original data according to $\mathbf{w}$. We implement such an adaptive sampling strategy in the function $\text{BEST}_h$ in Algorithm 2. The size of the subsampled dataset is controlled by a sampling ratio $\rho \in (0, 1]$, with the subsampled dataset of size $\lceil \rho n \rceil$.

By sampling the data adaptively, according to the weight vector $\mathbf{w}$, we sacrifice less accuracy than we would if we used static sampling. For instance, if the weight vector $\mathbf{w}$ in a given iteration of EXPGRAD$^{++}$ puts 90% of probability mass on 10% of examples, then picking the 10% examples with the largest weights results in a much smaller loss in accuracy than picking an arbitrary 10% of examples. This intuition can be turned into a formal argument showing that the adaptive sampling strategy yields an importance-weighted estimator of the objective $\sum_{i=1}^{n} w_i h(X_i)$ achieving the lowest variance among all importance-weighted estimators (see Appendix B for details). In our experiments, we show that adaptive sampling approach outperforms static sampling, and we do not see major losses in accuracy even when the sample size is only 25% or even less of the original dataset size (see Section 5).

## 5 Experimental evaluation

We next evaluate efficacy of our speedups. We first compare overall running times and quality of the solutions produced by EXPGRAD$^{++}$ with those of EXPGRAD, across a range of datasets, fairness constraints,

Table 1: Datasets used in experiments.

| Dataset | File size (MB) | #examples | #features | Sensitive attribute name | #values |
|---|---|---|---|---|---|
| Adult | 4.67 | $4.5 \times 10^4$ | 9 | Sex | 2 |
| COMPAS | 0.37 | $4.2 \times 10^3$ | 3 | Race | 2 |
| German | 0.05 | $1.0 \times 10^3$ | 9 | Sex | 2 |
| ACSEmployment | 319.47 | $3.2 \times 10^6$ | 99 | RAC1P | 9 |
| ACSPublicCoverage | 163.26 | $1.1 \times 10^6$ | 140 | RAC1P | 9 |

and learning oracles. We contextualize performance metrics by including additional baselines from fairness literature. We then dig into our two innovations separately, and conduct two ablation studies. In the first, we remove the sampling component of $\text{ExpGrad}^{++}$ and evaluate the impact of column generation on the number of oracle calls. In the second, we evaluate impact of sampling on the overall running time and solution quality.

### 5.1 Experimental setup

**Tasks and data.** We consider binary classification problems under two kinds of fairness constraints: demographic parity and equalized odds (Dwork et al., 2012; Hardt et al., 2016).

We consider 5 tabular datasets, whose main characteristics are shown in Table 1. Three of them (*Adult*, *COMPAS*, and *German*) are smaller datasets included in many previous studies (Islam et al., 2022a; Mehrabi et al., 2021; Pessach & Shmueli, 2023). Two larger datasets (*ACSEmployment* and *ACSPublicCoverage*s) have been proposed more recently (Ding et al., 2021) to enable larger-scale evaluations.

*Adult* (Becker & Kohavi, 1996) describes demographic and occupational attributes of several thousand individuals extracted from the 1994 US Census database. The task is to predict whether an individual has an income higher than 50K, with sex as the sensitive attribute.

*COMPAS* (Larson et al., 2016) contains arrest records, demographic information, and criminal history of defendants arrested in 2013–2014. The task is to predict reoffense within two years, with race as the sensitive attribute.

*German* (Hofmann, 1994) contains records of individuals applying for a loan. The task is to predict whether the default risk of an individual is high or low, using sex as the sensitive attribute.

*ACSPublicCoverage* and *ACSEmployment* (Ding et al., 2021) have been constructed from the US Census data collected within the American Community Survey. These datasets include information related to ancestry, citizenship, education, race, employment, language proficiency, income, disability, etc. The datasets differ in their prediction tasks: *ACSPublicCoverage* contains data for predicting whether an individual is covered by public health insurance, *ACSEmployment* contains data for predicting whether an individual is employed. We select race (RAC1P in the dataset) as the sensitive attribute. Unlike other datasets, in this case, the sensitive attribute is multi-valued.

**Models.** We run the reduction approach ($\text{ExpGrad}$ and $\text{ExpGrad}^{++}$) with two kinds of oracles: *LogisticRegression* and *HistGradientBoostingClassifier* from the *scikit-learn* library. Their hyperparameters are tuned separately for each dataset, without fairness constraints, using 5-fold crossvalidation. In the body of the paper, we report results only for logistic regression, and leave the results for gradient-boosted decision trees to the appendix.

For $\text{ExpGrad}$, we use the implementation available in the *fairlearn* library version 0.9.0 (Weerts et al., 2023). For $\text{ExpGrad}^{++}$, we augment the fairlearn implementation. In most experiments, we use the default settings of the optimization hyperparameters of $\text{ExpGrad}$. However, in some experiments we vary the learning rate

via the hyperparameter *eta0*, which specifies a multiplicative constant applied to the theoretical value of the learning rate. (The default value is *eta0=2.0*.)

In addition to EXPGRAD and EXPGRAD$^{++}$, we also consider an unmitigated approach (UNMITIGATED), which corresponds to running the base classification algorithm, and 6 additional methods from fairness literature discussed in Section 2:

- Two pre-processing approaches: CALMON (Calmon et al., 2017) and FELD (Feldman et al., 2015).
- Three in-training approaches: ZAFAR DI (Zafar et al., 2017b), ZAFAR EO (Zafar et al., 2017a), and FairGBM (Cruz et al., 2023).
- One post-processing approach: HARDT (Hardt et al., 2016).

Pre-processing and post-processing approaches are applicable to both demographic parity and equalized odds, and to both logistic regression and boosted tree models. ZAFAR DI is applicable to demographic parity, ZAFAR EO is applicable to equalized odds. Standard implementations of ZAFAR DI/EO[4] only support logistic regression models, so we only apply them to logistic regression. FairGBM is a method specifically designed for gradient boosted trees. While it is applicable to both demographic parity and equalized odds, its standard implementation[5] does not support demographic parity, so we only apply it to equalized odds.

We do not evaluate ZAFAR EO and CALMON on the two larger datasets since ZAFAR EO implementation does not support multi-valued sensitive attributes, and CALMON requires a problem-specific distortion function, which is not available for these datasets. Finally, note that HARDT requires access to the sensitive attribute at prediction time, which might not be available or allowed in some contexts.

The purpose of including these baselines is to provide context for the metric values appearing in the comparison of EXPGRAD$^{++}$ and EXPGRAD. We do not perform hyperparameter tuning for these baselines and just use the default settings provided by their implementations. However, we do consider several tradeoffs between accuracy and fairness for FairGBM and ZAFAR DI (see Appendix A). For more extensive cross-method comparisons, we refer the reader to bake-off papers like Islam et al. (2022a) (the reduction approach is evaluated in the appendix of its arXiv version, Islam et al., 2022b).

**Evaluation methodology and metrics.** All of our experiments were conducted on a machine equipped with Intel(R) Xeon(R) Platinum 8370C CPU @ 2.80GHz and 62.8 GB RAM, running on Ubuntu 20.04.5 LTS operating system. The accuracy is evaluated in terms of classification error. Fairness is evaluated using the DP difference $\Delta_{\text{DP}}$ and EO difference $\Delta_{\text{EO}}$, which were introduced in Section 3 (Eqs. 2 and 3) as a way to quantify the degree of violation of fairness constraints.

For each experimental configuration, we evaluate the performance of each algorithm using stratified 3-fold cross validation, executed twice with different random seeds, resulting in six replications of each experiment. Stratification is performed by jointly considering the sensitive attribute and the label.

## 5.2 Overall performance comparison

In our first set of experiments we compare EXPGRAD$^{++}$ and EXPGRAD in terms of running times and the quality of the solutions they return. In EXPGRAD$^{++}$, we use the sampling ratio $\rho = 0.25$ with the datasets *ACSPublicCoverage* and *ACSEmployment*, and do not use subsampling on the three smaller datasets. For both reduction algorithms, we consider several values of the constraint violation bound $\epsilon$ (see Eq. 4). Specifically, we consider $\epsilon \in \{0.001, 0.005, 0.01, 0.02, 0.05, 0.10, 0.15\}$, and evaluate the performance for each.

In Figure 1, we show the test error and training time as a function of test fairness violation, across 5 datasets and for two types of fairness constraints. EXPGRAD$^{++}$ and EXPGRAD are plotted as curves obtained from runs for different values of $\epsilon$, corresponding to different fairness–accuracy tradeoffs. For other methods (with the exception of ZAFAR DI and FairGBM in Appendix A), we just consider their default tradeoff parameter (where available), so they are plotted as points. For clarity, results are reported without error bars (plots with error bars are shown in Figure 4 in Appendix A).

---

[4]https://github.com/mbilalzafar/fair-classification
[5]https://github.com/feedzai/fairgbm

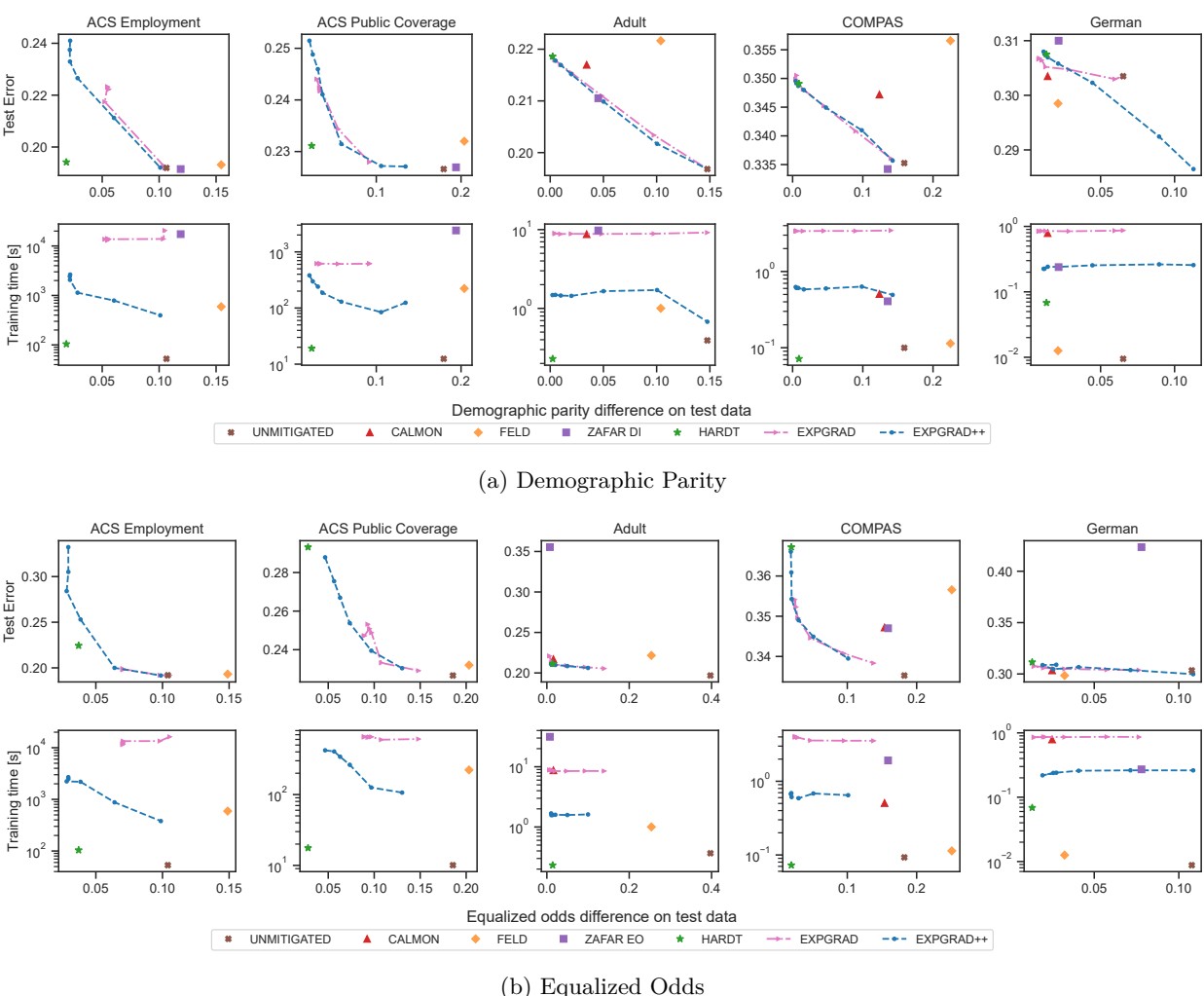

(a) Demographic Parity

(b) Equalized Odds

Figure 1: *Overall performance comparison (base learner: logistic regression).* Plotting test error and training time as a function of test fairness violation, averaged over 6 replicates. EXPGRAD$^{++}$ and EXPGRAD shown as curves, because they were evaluated at multiple fairness–accuracy tradeoff points; other methods shown as points. Results closer to lower-left corner of subplots are more favorable. Plots show that EXPGRAD$^{++}$ achieves the same fairness–accuracy tradeoffs as EXPGRAD, and matches or dominates the tradeoffs achieved by other methods except for HARDT, which however requires access to the sensitive attribute at inference time. EXPGRAD$^{++}$ successfully decreases the training runtime compared with EXPGRAD and matches the runtime of other methods except HARDT and UNMITIGATED (and FELD on the two smallest datasets).

We first focus on democratic parity results in Figure 1a. In the top row, we plot test error as a function of test fairness violation. Points towards lower left represent better models. Comparing EXPGRAD$^{++}$ and EXPGRAD, we see that EXPGRAD$^{++}$ can achieve the same fairness–accuracy tradeoffs as EXPGRAD, but it can also reach additional points along the tradeoff curve. In particular, EXPGRAD$^{++}$ is able to achieve lower values of fairness constraint violation than EXPGRAD, possibly because of better optimization. At the same time, as the second row shows, EXPGRAD$^{++}$ is substantially faster than EXPGRAD (up to a factor of 10 on large datasets).

Considering again the top row of Figure 1a, and now comparing EXPGRAD$^{++}$ against other baselines, we see that on the three small datasets, EXPGRAD$^{++}$ successfully matches or improves upon fairness–accuracy

tradeoffs achieved by other methods.[6] On the two larger datasets, the post-processing approach HARDT achieves a better fairness–accuracy tradeoff than $\textsc{ExpGrad}^{++}$. However, recall that HARDT requires access to the sensitive attribute at the inference time, which is not possible in many real-world scenarios. The second row of the figure shows that the unmitigated approach and post-processing are the fastest, whereas the running time of $\textsc{ExpGrad}^{++}$ is similar or better than the running times of the pre-processing and in-training methods (with the exception of FELD on the two smallest datasets: COMPAS and German).

The evaluation of equalized odds in Figure 1b yields analogous conclusions. Altogether these experiments show that $\textsc{ExpGrad}^{++}$ is able to speed up $\textsc{ExpGrad}$ by an order of magnitude, without sacrificing the quality of the resulting solutions. HARDT remains the best choice in settings where sensitive attribute is available (and is allowed to be used) at test time.

### 5.3 Efficacy of column generation

We next study the efficacy of column generation. We compare $\textsc{ExpGrad}$ with $\textsc{ExpGrad}^{++}$, but to isolate the effect of column generation, we do not perform any subsampling in $\textsc{ExpGrad}^{++}$. Since $\textsc{ExpGrad}$ is much slower than $\textsc{ExpGrad}^{++}$ (as we saw in Section 5.2), we limit this ablation study to the three smaller datasets: *Adult*, *COMPAS*, and *German*.

In both $\textsc{ExpGrad}$ and $\textsc{ExpGrad}^{++}$, we set the allowed constraint violation to $\epsilon = 0.005$. We run $\textsc{ExpGrad}^{++}$ with the default learning rate and default termination condition. For $\textsc{ExpGrad}$, we consider three different learning rates (specified in the *fairlearn* library via the hyperparameter `eta0` $\in \{0.5, 1.0, 2.0\}$), and for each, we take the best-performing solution among the three solutions (according to the duality gap). For $\textsc{ExpGrad}$, we only report the running time of the run with the best hyperparameter setting rather than the sum of all three runs—this gives an advantage to $\textsc{ExpGrad}$.

In Figure 2, we show how well the two algorithms optimize accuracy and fairness as a function of time. We focus on training metrics because these capture the progress of optimization. For $\textsc{ExpGrad}^{++}$, we plot the accuracy and fairness after reaching the termination condition based on the duality gap (since the algorithm always reaches the early termination condition), but for $\textsc{ExpGrad}$, we plot the performance at several different iterates, possibly corresponding to stopping before convergence condition is reached.

Figure 2 clearly shows that adding the column generation improves the efficiency of the original $\textsc{ExpGrad}$. The new algorithm, $\textsc{ExpGrad}^{++}$, reaches the solution of the same quality as eventually found by $\textsc{ExpGrad}$ in substantially fewer iterations, and hence with a substantially lower running time. For instance, focusing on the *Adult* dataset in Figure 2a, note that $\textsc{ExpGrad}$ starts with a solution that has a larger than desired constraint violation, and as the constraint violation decreases, the training error slightly decreases. The final solution, reached after 50 iterations, has the same fairness–accuracy tradeoff as reached by $\textsc{ExpGrad}^{++}$ after just 6 iterations.

### 5.4 Efficacy of sampling

Finally, we investigate how sampling impacts the running time and quality of solutions produced by $\textsc{ExpGrad}^{++}$. In all experiments, we set the allowed constraint violation to $\epsilon = 0.005$. We evaluate our adaptive sampling approach for the sampling ratios $\rho \in \{0.001, 0.004, 0.016, 0.063, 0.251\}$. As a baseline, we also consider static sampling, where the dataset is subsampled (uniformly at random) once at the beginning, and then $\textsc{ExpGrad}^{++}$ is run on the subsampled dataset (without any further subsampling). In both cases, we run $\textsc{ExpGrad}^{++}$ with column generation and a default setting of optimization hyperparameters. We also evaluate $\textsc{ExpGrad}$ (that is, the exponentiated gradient algorithm without column generation) with both adaptive and static sampling, and show performance of unmitigated approach with static sampling.

In Figure 3 we show how the running time, test error, and test constraint violation of the evaluated algorithms vary as a function of sampling ratio.

---

[6]On *German*, it appears that at low constraint violations, CALMON and FELD achieve a lower test error than $\textsc{ExpGrad}^{++}$, but the differences are not statistically significant (see plots with error bars in Figure 4a in Appendix A).

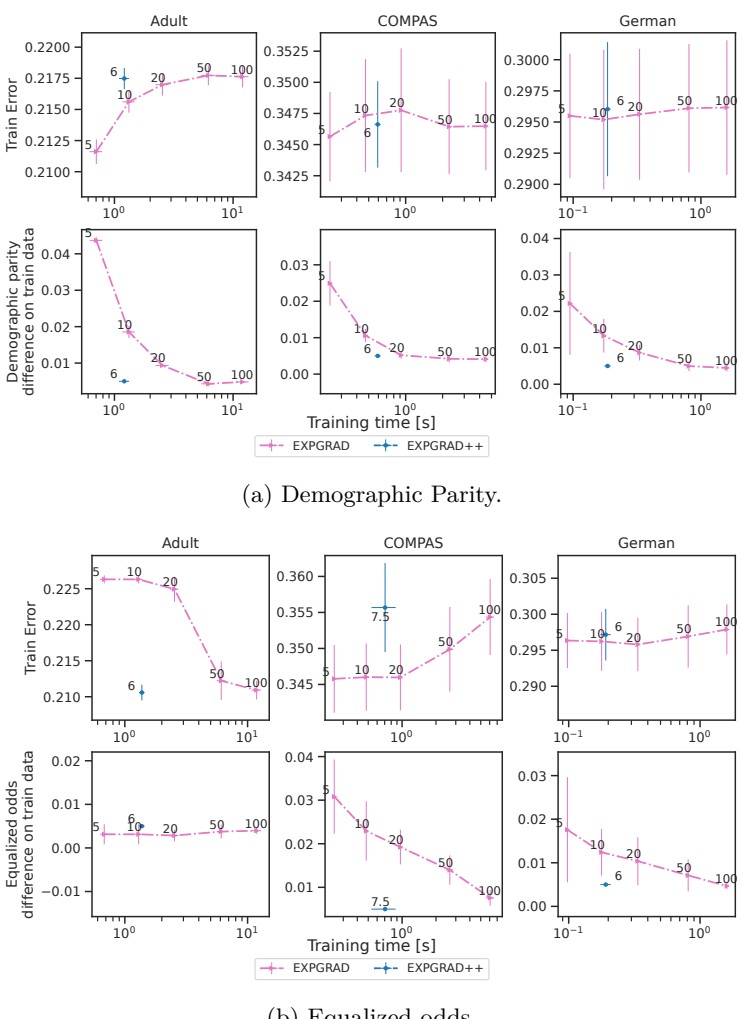

(a) Demographic Parity.

(b) Equalized odds.

Figure 2: *Efficacy of column generation (base learner: logistic regression).* Train error and fairness violation as a function of training time for 3 datasets, averaged over 6 replicates. For ExpGrad, showing performance at iterations ranging from 5 to 100 (indicated by numerical labels); for ExpGrad$^{++}$, showing performance at convergence (with a numerical label indicating the average number of iterations). ExpGrad$^{++}$ requires fewer iterations and thus less time to reach a solution of the same quality as ExpGrad.

Figure 3a shows results for demographic parity. The third row shows that ExpGrad$^{++}$ with both adaptive and static sampling always achieves the desired level of fairness. In contrast, ExpGrad with static sampling (and for *ACS Employment* also with adaptive sampling) fails to achieve the desired level of fairness. This is because severe subsampling makes the underlying optimization problems harder, so the algorithm fails to find a feasible solution within the default number of iterations $T = 50$ (this does not affect ExpGrad$^{++}$, which converges in all cases). Unmitigated solution, as expected, violates fairness constraint in all cases.

Continuing with the second row of Figure 3a, we see that using adaptive sampling allows both ExpGrad$^{++}$ and ExpGrad to achieve lower test errors compared with static sampling for the same sampling rates. In particular, note that adaptive sampling is able to achieve test error close to that of the entire dataset even at low sampling rates (0.01 for large datasets and 0.1 for small datasets).

Now focusing on the adaptive sampling variants only, we see in the second row of Figure 3a that on the three smaller datasets, both ExpGrad$^{++}$ and ExpGrad reach the same test error, but on the two larger datasets ExpGrad seems to be slightly better than ExpGrad$^{++}$. This however coincides with worse fairness due

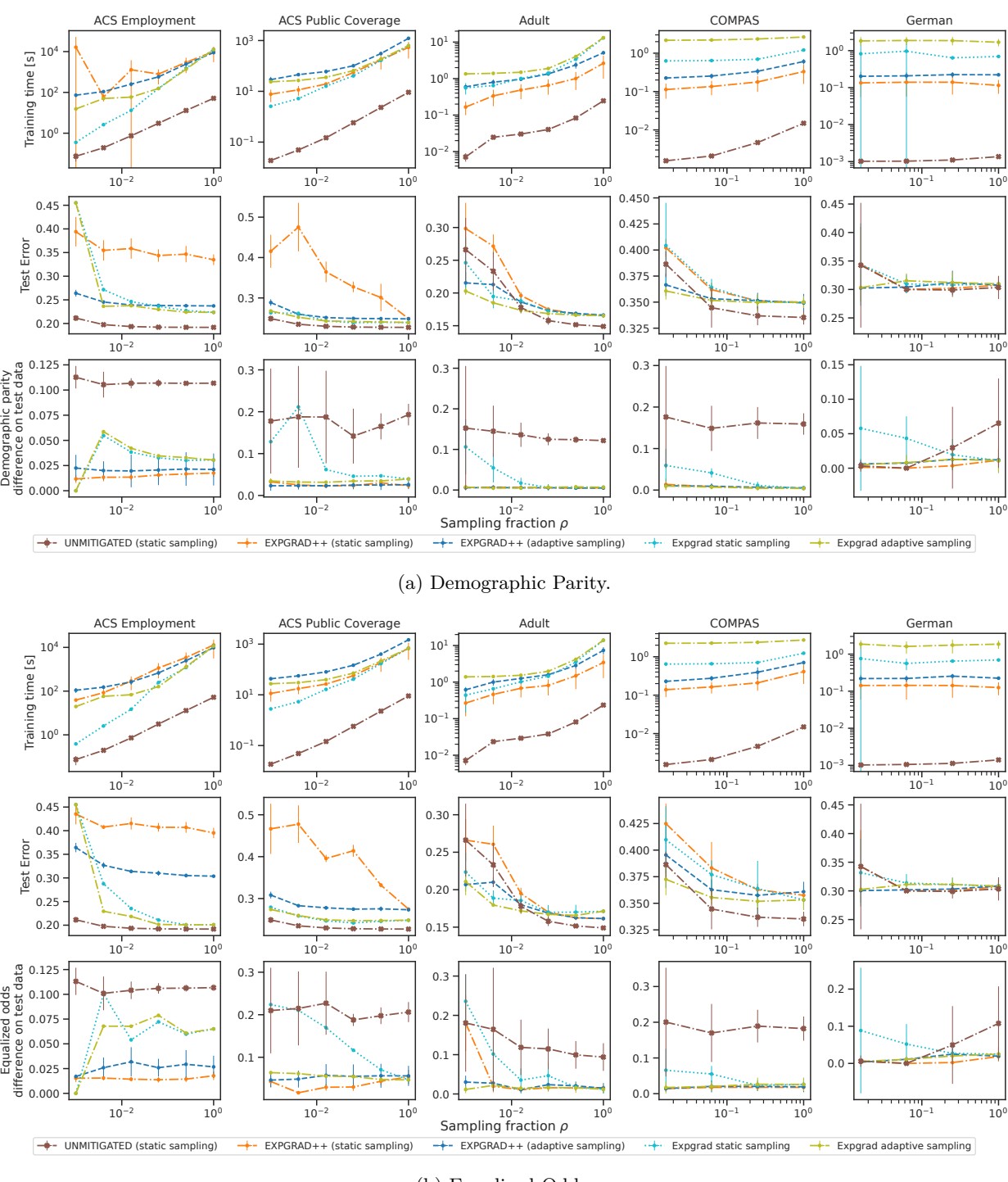

(a) Demographic Parity.

(b) Equalized Odds.

Figure 3: *Efficacy of sampling (base learner: logistic regression).* Plotting train runtime, test error, and test fairness violation as a function of sampling ratio; comparing adaptive and static sampling in EXPGRAD$^{++}$. Both adaptive and static sampling achieve low fairness violation, but adaptive sampling has lower test error. Ratios as low as 0.1 (or even less) yield improved runtime without sacrificing accuracy.

to the fact that EXPGRAD has not fully converged (in particular, see Figure 7 in Appendix A, showing that EXPGRAD optimization does not achieve the desired training fairness bound).

Finally, the first row of Figure 3a shows that subsampling improves running times, especially for larger datasets. Adaptive sampling is generally slightly slower than static sampling, as expected, because of the overhead of resampling at each iteration. One exception to this is *ACS Employment*, where EXPGRAD$^{++}$ with adaptive sampling is faster than EXPGRAD$^{++}$ with static sampling at low sampling ratios. This is because static subsampling undersamples small groups which leads to harder optimization problems, requiring more iterations to solve.

These general observations also carry over to the results for equalized odds in Figure 3b. Again, we see that adaptively selecting a small subsample of the dataset in EXPGRAD$^{++}$ (around 0.01 on larger datasets and 0.1 on smaller ones) yields substantial running time improvements without sacrificing the quality of the obtained solutions. The cases when EXPGRAD with adaptive sampling achieves better test error than EXPGRAD$^{++}$ with adaptive sampling coincide with cases when EXPGRAD has not achieve the desired training fairness (see Figure 8 in Appendix A), underscoring the importance of including column generation in the optimization procedure.

# 6 Conclusion

In this work, we have introduced two speedups in the reduction approach to fair classification: column generation and adaptive sampling. In our experiments on both small and large datasets, we have shown that the resulting algorithm EXPGRAD$^{++}$ matches the quality of solutions of the standard reduction algorithm (EXPGRAD), while substantially improving its runtime. As a result, reduction approach can be applied in a wider range of applications, including applications with larger datasets or more costly base algorithms.

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

# A    Additional experiment results

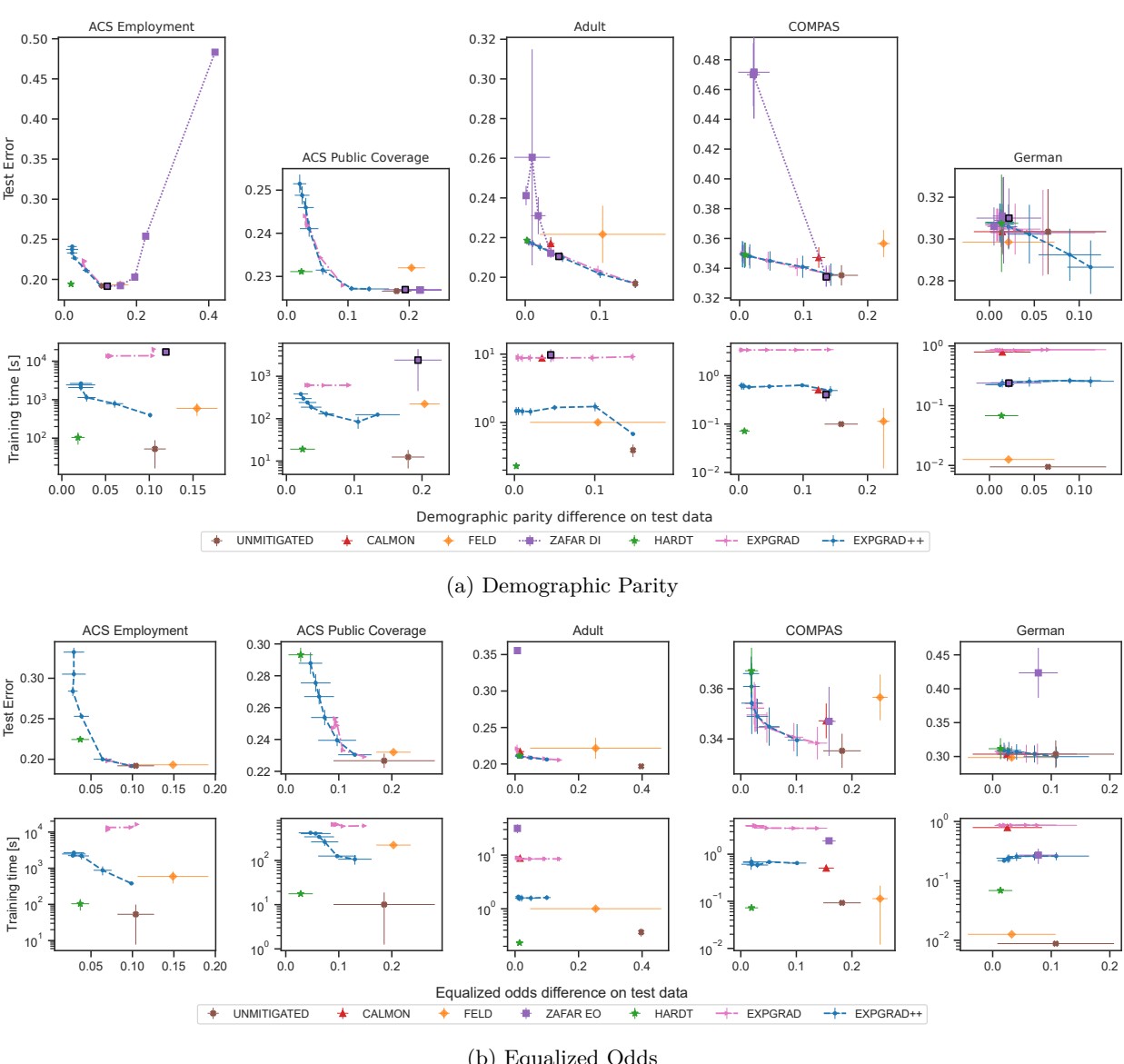

(a) Demographic Parity

(b) Equalized Odds

Figure 4: *Overall performance comparison (base learner: logistic regression).* For EXPGRAD, EXPGRAD$^{++}$ and ZAFAR DI we evaluated models corresponding to different fairness-accuracy tradeoffs. For ZAFAR DI, only one of the hyperparameter settings was trained on a comparable hardware with our other experiments, so only this comparable evaluation (designated with the solid black outline) is shown in the training time comparison in the second row. Several plots in the first row are taller to account for a larger scale of their y-axis compared with analogous test error plots on the same data sets.

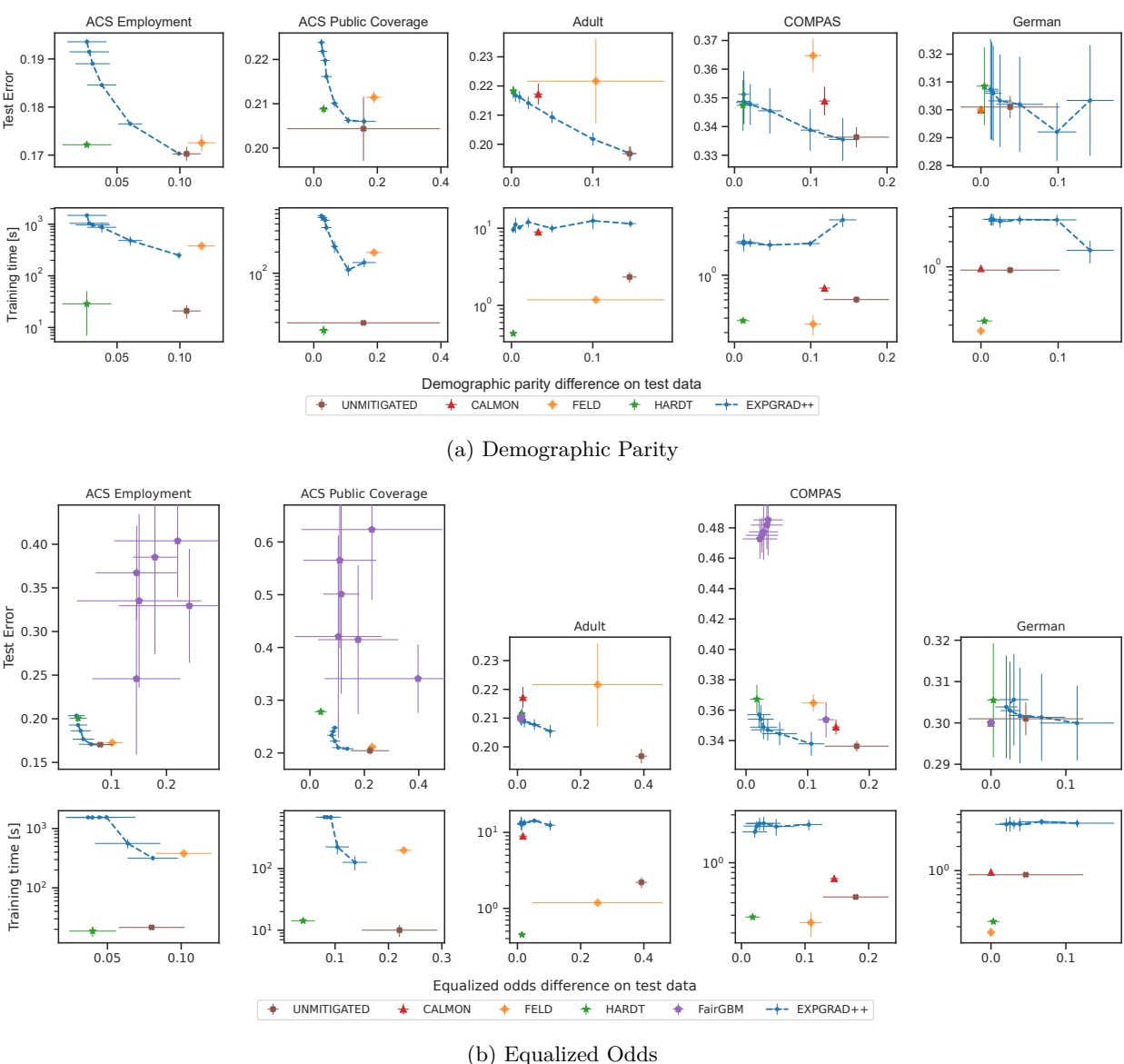

(a) Demographic Parity

(b) Equalized Odds

Figure 5: *Overall performance comparison (base learner: boosting).* For ExpGrad, ExpGrad$^{++}$ and FairGBM we evaluated models corresponding to different fairness-accuracy tradeoffs. FairGBM was evaluated on a different hardware configuration than our other experiments, so its training time is omitted in the last row. Several plots in the third row are taller to account for a larger scale of their y-axis compared with analogous test error plots on the same data sets.

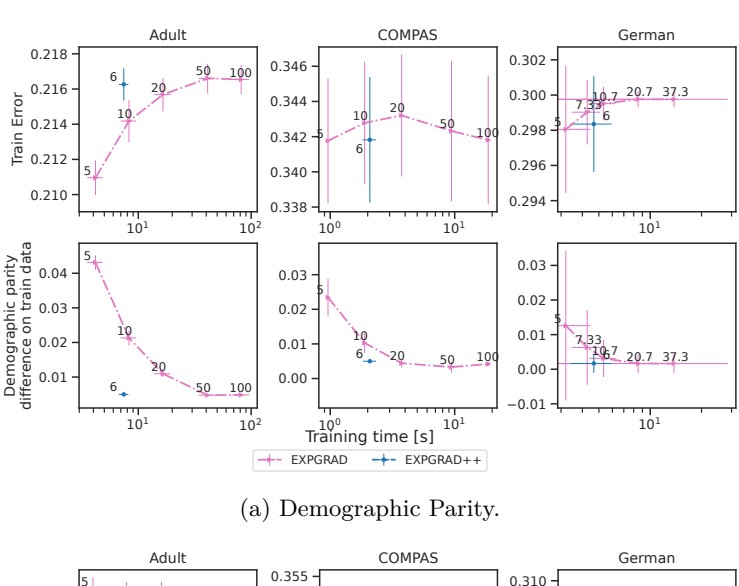

(a) Demographic Parity.

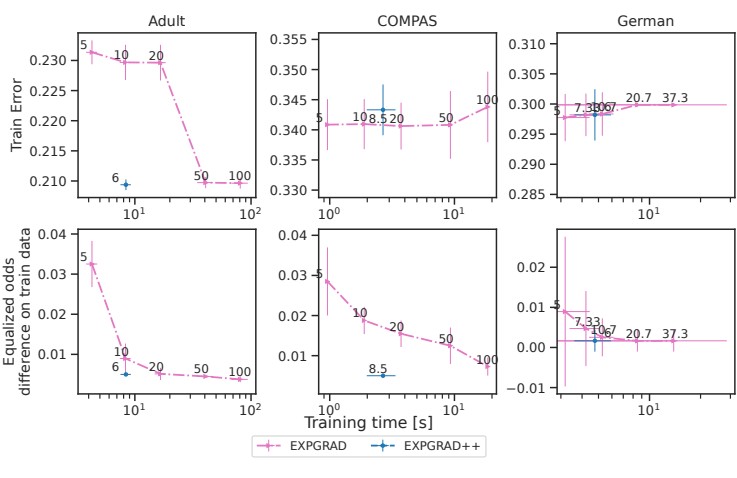

(b) Equalized odds.

Figure 6: *Efficacy of column generation (base learner: boosting).*

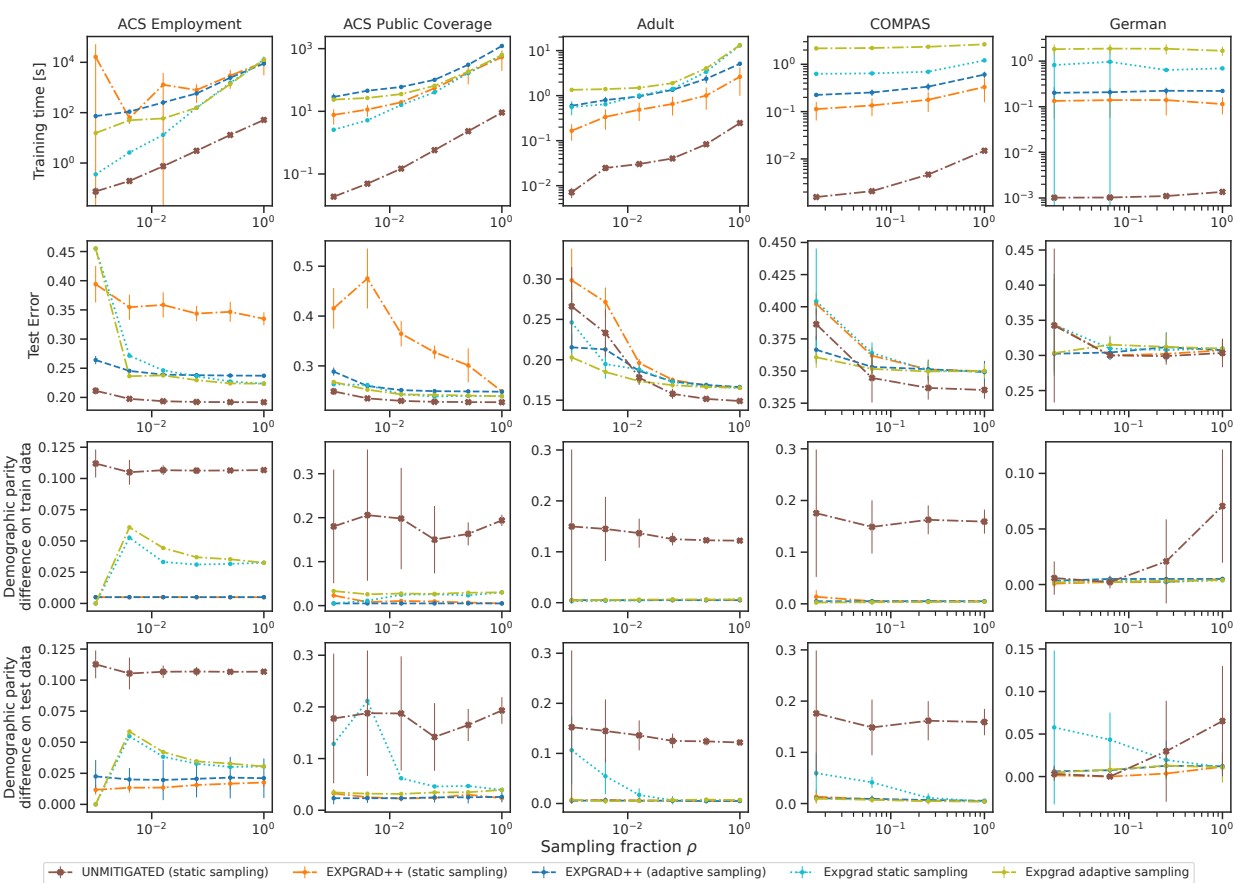

Figure 7: *Efficacy of sampling (base learner: logistic regression; fairness: demographic parity).*

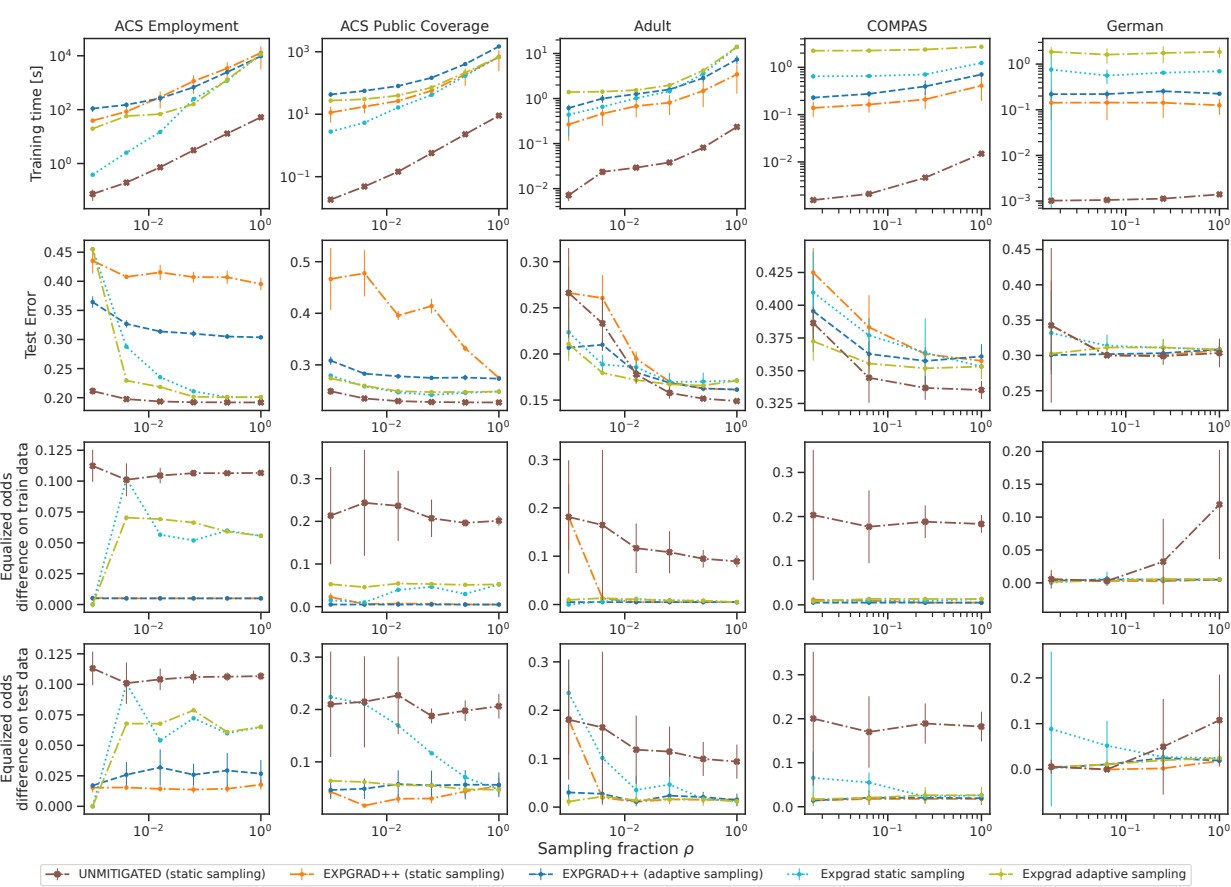

Figure 8: *Efficacy of sampling (base learner: logistic regression; fairness: equalized odds).*

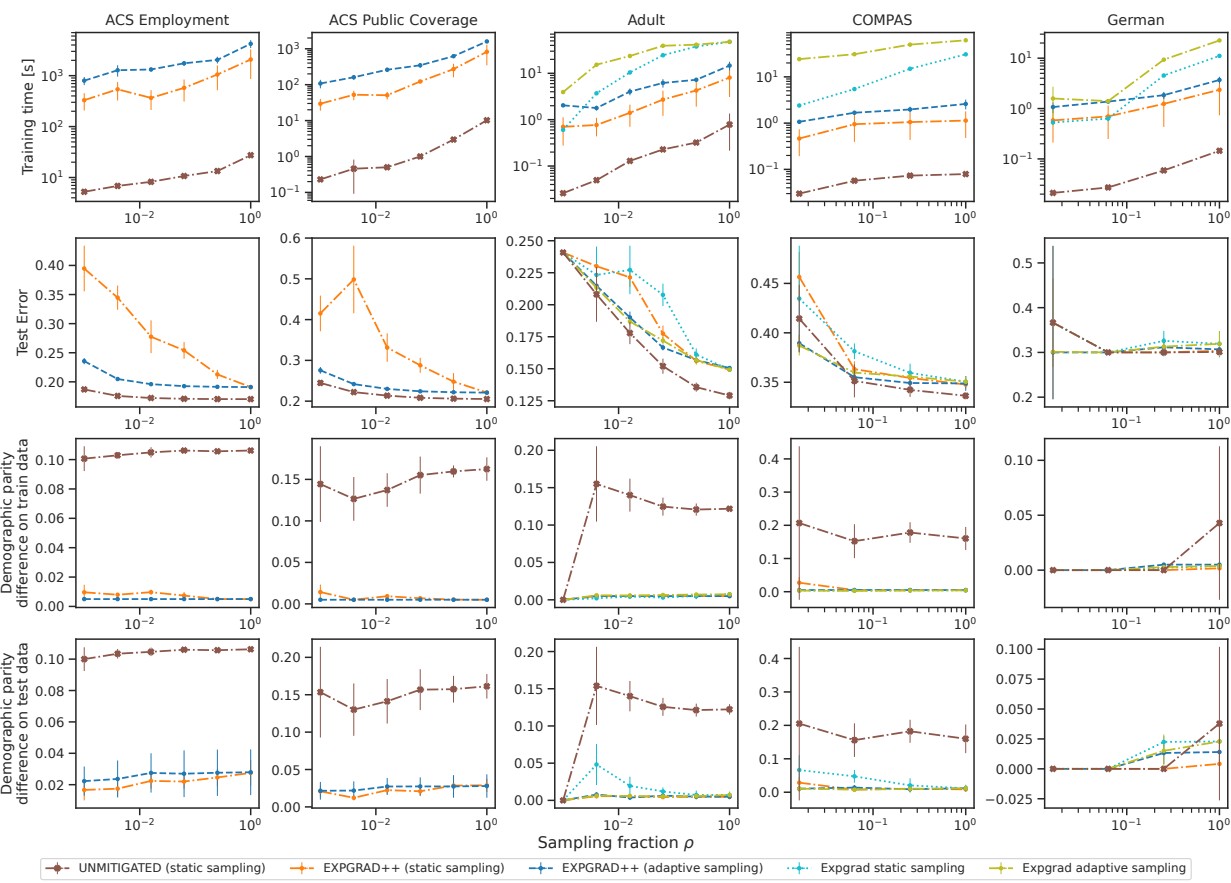

Figure 9: *Efficacy of sampling (base learner: boosting; fairness: demographic parity).*

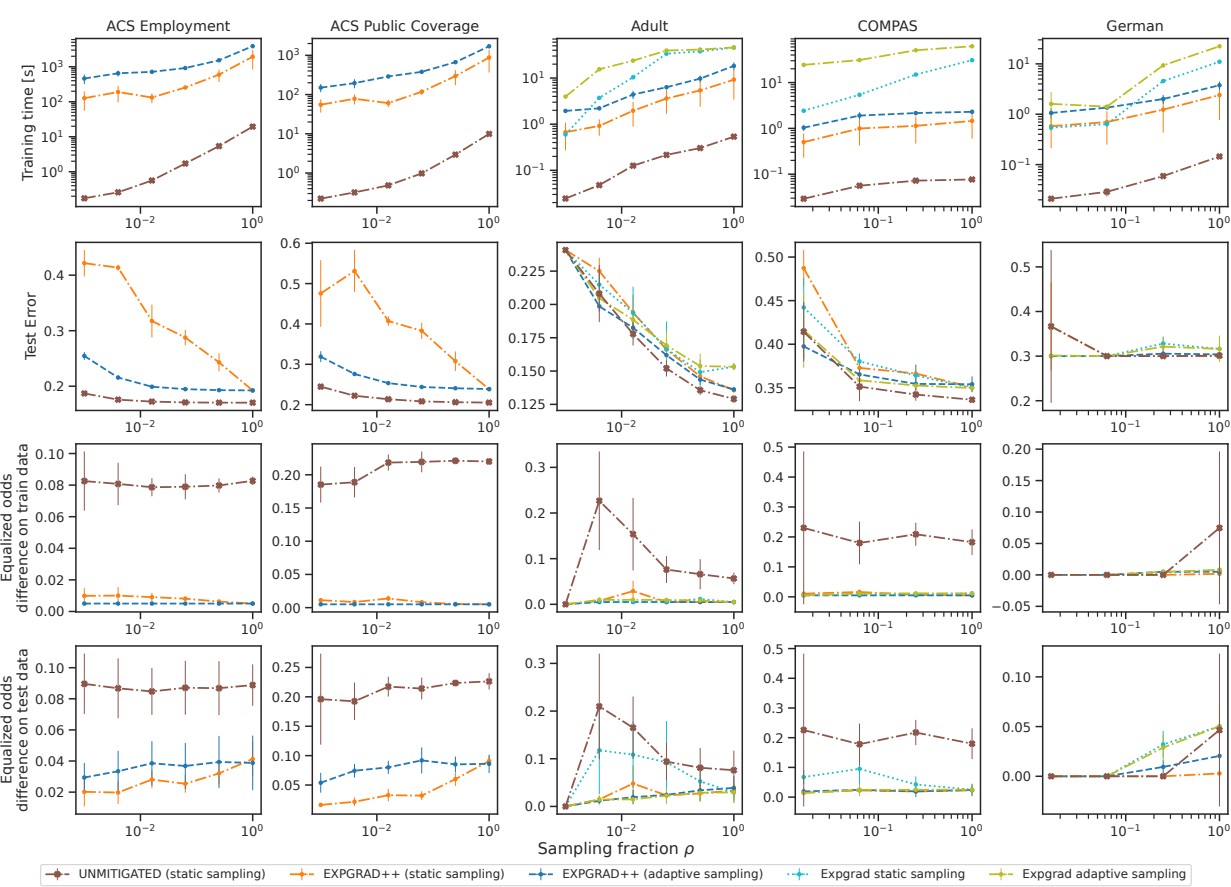

Figure 10: *Efficacy of sampling (base learner: boosting; fairness: equalized odds).*

## B   Importance sampling using weights $q_i$

In each call to $\text{BEST}_h$ in $\text{EXPGRAD}^{++}$, the algorithm first calculates the vector $\mathbf{w}$ and then seeks to find the classifier $h$ that approximately minimizes the sum

$$\sum_{i=1}^{n} w_i h(X_i), \tag{12}$$

which is the objective minimized by $\text{BEST}_h$ in $\text{EXPGRAD}$. In $\text{EXPGRAD}^{++}$, this objective is approximated using importance sampling with weights $q_i$. In this appendix we show that the weights $q_i$ used in $\text{EXPGRAD}^{++}$ correspond to the lowest-variance importance-weighted estimator of Eq. (12).

Assume that $w_i \neq 0$ for all $i$ (because indices $i$ with $w_i = 0$ can be dropped from Eq. (12) and they are also effectively ignored in the sampling carried out in $\text{BEST}_h$ in $\text{EXPGRAD}^{++}$ since they have $q_i = 0$). Furthermore, for our analysis, it is more convenient to consider minimization of a normalized and shifted objective

$$s(h) = \frac{1}{n} \sum_{i=1}^{n} w_i[2h(X_i) - 1] = \mathbb{E}_{i \in \text{Unif}(n)}\Big[w_i[2h(X_i) - 1]\Big],$$

where $\text{Unif}(n)$ refers to the uniform distribution over $\{1, \ldots, n\}$.

We approximate $s(h)$ using importance sampling (see, e.g., Section 3.3 of Robert & Casella, 2004). Specifically, assume that we are given importance weights $q_i > 0$, $\sum_{i=1}^{n} q_i = 1$. We sample $m$ indices $i_j$ for $j = 1, \ldots, m$ independently according to $\mathbf{q}$, and form the importance-weighted estimator

$$\hat{s}(h) = \frac{1}{m} \sum_{j=1}^{m} \frac{w_{i_j}}{n q_{i_j}}[2h(X_{i_j}) - 1]. \tag{13}$$

Let $Z_j = \frac{w_{i_j}}{n q_{i_j}}[2h(X_{i_j}) - 1]$ be the term corresponding to the $j$-th term in Eq. (13) and let $i' = i_j$. The expectation of $Z_j$ with respect to the random choice of $i'$ is then

$$\mathbb{E}[Z_j] = \mathbb{E}_{i' \sim \mathbf{q}}\left[\frac{w_{i'}}{n q_{i'}}[2h(X_{i'}) - 1]\right] = \sum_{i'=1}^{n} q_{i'} \frac{w_{i'}}{n q_{i'}}[2h(X_{i'}) - 1] = \frac{1}{n} \sum_{i'=1}^{n} w_{i'}[2h(X_{i'}) - 1] = s(h),$$

so $\hat{s}(h)$ is an unbiased estimator of $s(h)$.

The second moment of $Z_j$ can be bounded from below as

$$\mathbb{E}[Z_j^2] = \mathbb{E}_{i' \sim \mathbf{q}}\left[\left(\frac{w_{i'}}{n q_{i'}}\right)^2 [2h(X_{i'}) - 1]^2\right] = \mathbb{E}_{i' \sim \mathbf{q}}\left[\left(\frac{w_{i'}}{n q_{i'}}\right)^2\right]$$

$$= \sum_{i'=1}^{n} q_{i'}\left(\frac{w_{i'}}{n q_{i'}}\right)^2 = \left[\sum_{i'=1}^{n} q_{i'}\left(\frac{w_{i'}}{n q_{i'}}\right)^2\right] \cdot \left[\sum_{i'=1}^{n} q_{i'}\right]$$

$$\geq \left[\sum_{i'=1}^{n}\left(\sqrt{q_{i'}}\frac{|w_{i'}|}{n q_{i'}}\right) \cdot \sqrt{q_{i'}}\right]^2 = \left[\frac{1}{n}\sum_{i'=1}^{n}|w_{i'}|\right]^2. \tag{14}$$

The second equality is because $h(X_{i'}) \in \{0, 1\}$. The fourth equality is because $\sum_{i'=1}^{n} q_{i'} = 1$, and the inequality follows by the Cauchy-Schwarz inequality. Thus, the variance of $Z_j$ is bounded below by

$$\text{Var}[Z_j] = \mathbb{E}[Z_j^2] - (\mathbb{E}[Z_j])^2 \geq \left[\frac{1}{n}\sum_{i'=1}^{n}|w_{i'}|\right]^2 - s(h)^2,$$

with the lower bound achieved when the Cauchy-Schwarz inequality in Eq. (14) holds with equality. This occurs when

$$\sqrt{q_i} = c\left(\sqrt{q_i}\frac{|w_i|}{n q_i}\right)$$

for some constant $c$. Rearranging, this is equivalent to $q_i = c|w_i|/n$ for some constant $c$; in particular, this is achieved by setting $q_i = \frac{|w_i|}{\sum_{i'=1}^{n}|w_{i'}|}$. Thus, importance weights $q_i$ used in $\text{ExpGrad}^{++}$ give rise to the lowest-variance importance-weighted estimator of $s(h)$.

Let $W = \sum_{i=1}^{n}|w_i|$, so we have $q_i = |w_i|/W$ for all $i$. For this choice of $q_i$, the term $Z_j$ in the estimator $\hat{s}(h)$ becomes

$$Z_j = \frac{w_{i_j}}{nq_{i_j}}\left[2h(X_{i_j}) - 1\right] = \frac{W}{n} \cdot \frac{w_{i_j}}{|w_{i_j}|} \cdot \left[2h(X_{i_j}) - 1\right]$$

$$= \frac{W}{n} \cdot \text{sgn}(w_{i_j}) \cdot \left[2h(X_{i_j}) - 1\right] = \frac{W}{n} \cdot \left[1 - 2Y'_{i_j}\right] \cdot \left[2h(X_{i_j}) - 1\right] = \frac{W}{n} \cdot \left[2 \cdot 1\{Y'_{i_j} \neq h(X_{i_j})\} - 1\right],$$

where $Y'_{i_j} = 1\{w_i < 0\}$ is the label associated with the $j$-th sampled index in $\text{Best}_h$ in $\text{ExpGrad}^{++}$. Thus,

$$\hat{s}(h) = \frac{1}{m}\sum_{j=1}^{m} Z_j = \frac{W}{nm}\sum_{j=1}^{m}\left[2 \cdot 1\{Y'_{i_j} \neq h(X_{i_j})\} - 1\right].$$

So minimizing $\hat{s}(h)$ is equivalent to minimizing

$$\sum_{j=1}^{m} 1\{Y'_{i_j} \neq h(X_{i_j})\},$$

which is precisely the classification problem passed to the base classification algorithm in $\text{Best}_h$ in $\text{ExpGrad}^{++}$.

