# OpenReview forum: "Speeding up fairness reductions"
_TMLR — Accepted by TMLR_

### Review · Reviewer_Qr6m · 2026-03-10

**Summary Of Contributions:**

The authors suggest two computational improvements of Agarwal et al.'s (2018) ExpGrad algorithm. First, they suggest using a supplementary column generation step that precisely solves the classification problem under a fairness constraint within a smaller hypothesis space. Second, they suggest using subsampling of training data during oracle calls. Together, these improvements reduce the time required for max-min optimization by an order of magnitude on a selected set of tabular datasets.

**Strengths**
1. The problem is reasonable. The approach seems to bring computational benefits.
2. The application of tweaks is novel.
3. The writing is clear.

**Weaknesses**
1. The paper focuses on an in-processing constraint-based approach, which is computationally expensive for big neural networks.
2. While the proposed tweaks bring computational benefits, they are not novel.
3. I think the discussion in Section 4.2 is conceptually flawed. The authors suggest sampling points with the highest fairness violations. However, this suggestion might lead to a loss of diversity (e.g., because violations for majority groups are small). This loss of diversity might be detrimental for data with complex non-linear relationships. While Section 5.4 shows the superiority of the proposed sampling scheme, it uses a static baseline for comparison. The use of a static baseline seems unfair, since the adaptive baseline "uses" more unique points during training. A more appropriate comparison would be between the proposed dynamic procedure and the uniform dynamic procedure.

**Audience:**

Yes

**Audience Explanation:**

The paper might be of interest to people who engage with fair classification for tabular datasets. At the same time, the significance of the work seems somewhat incremental to me.

**Claims And Evidence:**

Yes

**Claims Explanation:**

Generally, I think the bulk of the evidence is sufficient to show that the proposed procedure is not worse than the original one and, at the same time, faster.
1. To strengthen the results, the authors might add a comparison with a different sampling baseline in Section 5.4, as I discussed above.
2. Additionally, the authors should consider a comparison with the FairGBM method (Cruz et al., 2023).

Cruz, A., Belém, C. G., Bravo, J., Saleiro, P., & Bizarro, P. (2023). FairGBM: Gradient Boosting with Fairness Constraints.

**Requested Changes:**

1. Please address my concern about Section 4.2.
2. Please add experiments with FairGBM and the alternative sampling scheme described above.

---

> ### Author Response · Authors · 2026-03-31
> **response to the points raised**
>
> Thanks for the thoughtful review. Below we respond to the points you've raised.
>
> **(3A) Not applicable to large neural networks**
>
> That's correct. Even 10 calls of the base learner might be too expensive when the base learner is a large language model. We will mention this in our limitations section.
>
> **(3B) Discussion in Section 4.2 is conceptually flawed**
>
> We do not think that there is a conceptual flaw in our reasoning, but it is true that we have just provided informal motivation, so there is space to make things clearer.
>
> There are two issues at play here. First, note that the final solution is obtained by randomizing across classifiers obtained in all iterations of exponentiated gradient, and these classifiers have been trained on data sampled from different distributions (data points that violate constraints in one step are not necessarily the same as data points that violate constraints in the next step), so while each individual distribution might appear like it is losing some diversity, we do not lose diversity overall.
>
> Second, the distribution from which we sample in each iteration is chosen to provide an unbiased and **low variance** estimate of the objective in that iteration. More precisely, based on the Lagrange multipliers in a given iteration, we form weights $w_i$, which give rise to the objective of the form $S=1/n\sum_{i=1}^n w_i h(X_i)$. The goal of the call to the base learner in that iteration is to minimize $S$. For any fixed $h$, we can view $S$ as the expected value of the quantity $Z_i=w_i h(X_i)$ if $i$ is sampled uniformly at random from $\\{1,\dots,n\\}$. To approximate the value of $S=\mathbb{E}_{i\sim uniform}[Z_i]$, we use importance sampling (see, e.g., https://en.wikipedia.org/wiki/Importance_sampling#Application_to_simulation, or Section 3.3.2 of [Robert-Casella, 2004](https://link.springer.com/book/10.1007/978-1-4757-4145-2)). The importance weights that give rise to the lowest-variance estimator are of the form $p_i=|Z_i|/c$ where $c$ is the normalizing constant. Of course, the sampling probabilities $p_i$ need to work across many different classifiers $h$. Depending on the value of $h(X_i)$ we have either $Z_i=0$ or $Z_i=w_i$. From the perspective of variance, we make a pessimistic assumption that $Z_i=w_i$ (thus leading to a larger possible variance), which gives rise to the sampling distribution $p_i=w_i/c$, introduced in Section 4.2. The motivation that we provide there is meant to communicate the basic intuition behind importance sampling: that terms with a larger potential magnitude should be sampled with proportionally larger probabilities, because they can contribute more to the objective.
>
> The reasoning in the previous paragraph is somewhat heuristic, but when the set of classifiers $h$ is reasonably expressive then we indeed expect that we need to account for the magnitude of each $|Z_i|$ of up to $|w_i|$.
>
> We hope that this clarifies our choice of the sampling distribution. We will add some of these details to the manuscript.
>
> **(3C) Add uniform (but fresh-in-each-iteration) sampling baseline**
>
> Based on the reasoning in our answer (3B) above, we expect uniform sampling (even one that is done independently in each iteration) to have a larger variance than importance sampling. This means that the objective passed to oracle calls has more noise added to it (compared with the data obtained by importance sampling) and thus yields worse classifiers (the addition of noise can be viewed as underfitting).
>
> Adding the baseline that applies fresh uniform sampling in each iteration as requested by the reviewer is a somewhat more substantive request (since it affects quite a few experiments), and so we would only consider it if the reviewer still views this baseline as necessary to properly assess our contribution (after seeing our explanation in point (3B) above). Please also see our general comments regarding adding more experiments.
>
> **(3D) Add comparsion to FairGBM**
>
> This is a reasonable request, and we will look into this. However, after perusing the documentation of the Python package referenced in the FairGBM paper, it is not clear whether demographic parity is supported out of the box, so we might only have these experiments for equalized odds. Moreover, we will not be able to obtain comparable wall clock times (see our general comments regarding adding more experiments). As such, the only material update would be in Appendix A, Figure 5b, top row. Let us know if you view this as worthwhile.

---

> > ### Comment · Reviewer_Qr6m · 2026-04-06
> >
> > Thank the authors for their response.
> >
> > 1. Given the constraints on new experiments, I do not insist on adding them for the current submission. However, I still think that both experiments will strengthen the paper.
> > 2. As for Section 4.2, I still think that it is conceptually flawed. While I agree that subsampling is reasonable for the narrow task of optimizing objective (8) at any particular step, this narrow goal is not fully aligned with the broader goal of training a fair generalizable classifier. First and most crucially, such subsampling might hurt the generalization since the empirical objective is not equivalent to the test risk. This discrepancy is the reason behind my initial concern about the diversity of points. Second, the current implementation of importance weighing seems incorrect. Line 23 in Algorithm 2 minimizes the misclassification error. However, the sign of $c_i$ might in theory be different from the sign of $w_i$, making the classification problem different from the fair learning problem. Third, while the authors claim that their weighting scheme "minimizes" the objective's variance, this assertion is not entirely clear-cut because the penalties, and hence the objective, evolve over time (but for this specific point, I agree that the approach is at least reasonable).

---

> ### Author Response · Authors · 2026-04-06
> **follow-ups regarding Section 4.2**
>
> Regarding overfitting (and mismatch between training and test loss), we generally follow a similar strategy as in the original papers that introduced reductions (see the footnote preceding Eq. 1): the complexity of the classifiers is controlled via appropriate selection of the hyperparameters of the base model. This allows the reduction approach to focus on the optimization problem without worrying about overfitting.
>
> Because of the concerns about correctness, we have added Appendix B. In that appendix, we formalize our reasoning from the previous reply to show that our adaptive sampling always yields the lowest-variance importance-weighted estimator of the objective in a given iteration. We have also explicitly derived the classification problem passed on to the base classifier, starting with the weights $w_i$ and showing how the labels $Y$ of the sampled data points are determined from $w_i$. We hope this addresses the concern about whether we have an unbiased approximation of the objective $\sum_{i=1}^n w_i h(X_i)$.
>
> Generally, we agree with the reviewer that our arguments (although precise and correct) are somewhat idealized. Similar to the original reductions papers, our reasoning assumes that the oracle returns a classifier from $\cal{H}$ that minimizes the empirical classification error on the provided (sampled) dataset. In practice, the set $\cal{H}$ is often specified implicitly (via a penalty function) and the base classification algorithm is minimizing some convex upper bound on the classification error (like logistic loss).

---

> > ### Author Response · Authors · 2026-04-10
> > **comparison with FairGBM added**
> >
> > FYI: We have just uploaded a revision with FairGBM experiments added in Appendix A, Figure 5b.

---

### Review · Reviewer_7CJD · 2026-03-20

**Summary Of Contributions:**

The main contribution of this paper is to improve the efficiency of the _reduction approach_ of ExpGrad for fair classification. The paper proposes ExpGrad++ as a more efficient approach based on column generation reducing the number of oracle calls needed and an adaptive subsampling of training samples per oracle call to reduce the cost of each oracle call. Together, both optimization tricks yield roughly a 10x speedup on standard fairness benchmarks without degrading the fairness-accuracy tradeoff.

### References used in this review
1. FairGBM (ICLR 2023): https://openreview.net/forum?id=x-mXzBgCX3a
2. Fair-AutoML (ESEC/FSE 2023): https://dl.acm.org/doi/10.1145/3611643.3616257
3. Fairness-aware AutoML survey (arXiv 2023): https://arxiv.org/abs/2303.08485
4. Fair ensembles (AutoML 2023): https://proceedings.mlr.press/v224/feffer23a
5. Constrained stochastic optimization (Optimization Letters, 2024): https://link.springer.com/article/10.1007/s11590-023-02024-6
6. FairMixup (ICLR 2021): https://openreview.net/forum?id=DNl5s5BXeBn

**Audience:**

Yes

**Audience Explanation:**

- **More Efficient Computation:** The paper improves the efficiency of a well established method in fairness classification. The improvements are, in my opinion, an engineering feat which have, however, been done with a very high quality. The paper addresses a limitation of the current state of the art and improves on it. This improvement is rather incremental but nonetheless important and I am sure the improvements will already help the community, particularly if implemented in _fairlearn_.
- **Practical relevance:**. The reduction approach is already implemented in _fairlearn_, which is one of the most widely used open-source fairness toolkit. The paper's modifications should be backward-compatible as ExpGrad++ is a drop-in replacement that would preserve the same API and the same theoretical guarantees while running faster. This means the contribution should realistically be merged into the library and immediately benefit practitioners.

**Broader Impact Concerns:**

The paper deals with fair machine learning. In my opinion any work in this field should have a broader impact statement which it does not yet have.

**Claims And Evidence:**

Yes

**Claims Explanation:**

- **Clearly written:** The paper is very clearly written and presented it has a high quality. Algorithm 2 (with 1) clearly shows how the contribution adds to the existing methods.
- **Contribution and Improvements are not oversold:** The paper does not oversell its contribution. While the contribution is **more incremental and engineering-sided**, the paper clearly states it and does not market itself as something different.
- **Reproducibility.** The paper builds on _fairlearn_ (a public fair ml library) and uses publicly available datasets.  All hyperparameter settings are specified. The paper also describes the hardware used. Someone could easily reproduce these results without needing to email the authors, which is much more than can be said for a lot of published work.
- **Empirical evaluation:** The paper's empirical evaluation is based on five datasets spanning some orders of magnitude in size (1K to 3.2M samples), two fairness definitions, two model families, and seven values of the constraint violation bound $\epsilon$. This already spans quite a good empirical setup. While I think the baselines are quite dated (see weakness below), in total, six baselines covering all three intervention categories (pre-processing, in-training, post-processing) are evaluated.

### Minor Concern
- **Limitation:** The paper does not have a dedicated limitations section, which in my opinion should be part of any paper accepted at TMLR.

**Requested Changes:**

- **More timely comparisons**. The empirical evaluation of this work feels like it was done about ten years ago. All baselines are from 2015 to 2017. The fairness field has changed after this with AutoML frameworks incorporating fairness or direct fairness optimization in gradient boosting and optimization techniques (see References [1-6]). While those works do not necessarily have to be compared against for the contribution here, I think its a striking signal that the work done here is not so timely and foundational. A comparison with some more modern approaches would in my view substantially improve this paper.
- **More interesting data sets and models**. The paper only considers 5, in my option rather dated, data sets and only two model classes (logistic regression and gradient boosted trees) for the classification task. I am sure the results of the paper would also translate to more interesting models (deep learning) where runtime is actually of more importance than the here used model families.

I want to stress that, **both changes are not critical to secure my recommendation it would simply strengthen the work in my view.**

---

> ### Author Response · Authors · 2026-03-31
> **response to the points raised**
>
> Thanks for a thoughtful review. Here are our responses to your concerns and suggestions:
>
> **(2A) Limitation section / impact statement missing**
>
> Thanks for calling these out. We will add sections on recommended uses, risks and limitations. Our preferred choice is to mention these in the introduction (rather than in a section at the end of the paper), but let us know if you prefer otherwise.
>
> **(2B) Lack of more timely baselines**
>
> This is a reasonable concern. We made a conscious decision to work with some of the highest cited baselines that appear in many cross-method comparisons, but that obviously biased us towards older methods. Based on this concern and also the concern raised by Reviewer Qr6m (see our response to the point 3D), we will look into incorporating FairGBM (with equalized odds) as an additional in-processing baseline. (However, please also see our general comments regarding adding more experiments.)
>
> We will also add a discussion of AutoML approaches as a natural extension of direct optimization techniques in the related work section, and we will add a discussion of how our work relates to fair ensembles: namely, reduction approach can be viewed as a specific form of ensembling, and so it can be incorporated in the framework of Feffer et al. 2023.
>
> **(2C) More interesting data sets/models**
>
> This is a valid point and we'll include it in the limitations section.

---

> > ### Comment · Reviewer_7CJD · 2026-04-01
> > **Acknowledging the Response**
> >
> > **I thank the authors for their response to my review!** As I already said the points raised are no deal-breakers for me. I actually really like your idea for moving the limitation section in the introduction. Please, do not just add in the paragraphs but give it named section/paragraph headings. This makes it so much easier to find it. I think this discussion about the datasets/methods are important and help to position the paper. Particularly, if no additional experimental results will be obtained for some, now important, methods. I also think the FairGBM results would be very welcome! In summary, I am still happy overall with the submission which considerably improves on the current state-of-the-art in some cases.

---

> > > ### Author Response · Authors · 2026-04-10
> > > **comparison with FairGBM added**
> > >
> > > FYI: We have just uploaded a revision with FairGBM experiments added in Appendix A, Figure 5b.

---

### Review · Reviewer_9xny · 2026-03-23

**Summary Of Contributions:**

To train a fairness-aware classifier, the "reductions approach" is known that it can accept various training methods and fairness criteria. However, it requires many times (typically 100) of base training computations ('base' intends a method that do not consider the fairness), which makes the method very costly. During the procedure, it iteratively repeats the optimization of the classifier and $\\lambda$, where $\\lambda$ is optimized to satisfy the constraint of unfairness being under given threshold. The proposed method achieved faster fairness-aware training by introducing additional convergence check as follows: instead of checking the objective function (in this case, Lagrangian function) directly, we consider a minified objective function using only the classifiers examined so far (cf. column generation method). Also, instead of the weighted training in the existing method, the proposed method subsamples the training samples in the weight, to reduce the computational cost. Experimentally, it made the proposed method by about 10 times faster while retaining the trade-off performance between the prediction and the fairness.

**Audience:**

Yes

**Audience Explanation:**

Fairness in machine learning becomes more and more important, and the base method is known to be versatile (various training methods and fairness constraints are accepted). So improving the cost of such method looks valuable.

**Broader Impact Concerns:**

Although methods to achieve fairness need careful consideration in real-life uses, it looks that the proposed method itself does not cause fairness issues since the training is conducted under fairness constraints.

**Claims And Evidence:**

No

**Claims Explanation:**

Most parts are fine and clear with high quality, but several parts may be essential defects. Especially, experimental comparisons with ZAFAR methods may be needed to be reexamined (detailed in "Requested Changes" part).

**Requested Changes:**

## Points critical to securing my recommendation for acceptance

- Overall: The reviewer felt that the effect of the column generation emerged *because* we train a randomized classifier. If we do not train a randomized classifier but a deterministic classifier, does the column generation still have an effect of cost reduction?
  - If not, it is desired to compare the proposed method with the existing method with deterministic classifier.
- Section 4 (overall): What can be the reason why this problem can be made very fast by the column generation method? Since not all (high-dimensional) optimization problems can be made faster by the column generation method, so the problem is expected to have a property good for the column generation method.
- Section 4.2 (Line 22 in Algorithm 2): In the proposed adaptive sampling, why $Y$ is set to $1\\{ w\_i < 0 \\}$ rather than $Y\_i$? Why is it valid?
- Section 5.1: As far as reading the papers of ZAFAR DI/EO, it is true that ZAFAR EO is applicable only to the logistic regression and similar, but ZAFAR DI is not limited to the logistic regression. Also, since both ZAFAR EO and ZAFAR DI have user-specifiable parameters of the fairness, it looks that we can "draw" lines of fairness/accuracy trade-off for these methods like the proposed ones. If my understanding is correct, please consider setting up experiments for these issues.

## Points that simply strengthen the work in my view

- Title: The title "fairness reduction" looks that the meaning of "reducing the fairness" (i.e., the term "reduction" looks a general term). I suggest the title "Speeding up reduction-based fairness".
- Section 3.2: Although the paper states that "See Agarwal et al. (2018) for full derivation of $\\mathbf{A}$ for DP, EO, as well as for more general fairness constraints", but I suggest that the reference is used only for the derivations of $\\mathbf{A}$, $\\mathbf{b}$, $\\mathbf{c}$ and $c\_0$; the expressions of $\\mathbf{A}$, $\\mathbf{b}$, $\\mathbf{c}$ and $c\_0$ themselves should be presented in this paper (main text or appendix), since they are needed to implement the proposed method.

---

> ### Author Response · Authors · 2026-03-31
> **response to the points raised**
>
> Thanks for the thoughtful review. Below we respond to the critical points raised in the review as well as the points towards strengthening the manuscript (paraphrased):
>
> **(1A) Overall: Could you run column generation with deterministic classifiers?**
>
> The reviewer is correct that column generation is only applicable in specific settings: namely, if the problem being solved is a linear program. The focus of our paper is on speeding up the reduction approach (the algorithm ExpGrad), which is formulated as a linear program with an intractably large set of columns, and hence column generation is a suitable strategy. It is also true that the reduction approach is designed to return a randomized classifier, which is expressed in the linear problem solved. We are not sure what it would mean to evaluate column generation with a deterministic classifier. (What linear program would be solved?)
>
> Note that we do not claim that our two improvements (column generation and sampling) are applicable or effective for arbitrary classification algorithms just for ExpGrad.
>
> **(1B) Section 4 (overall): Why/when does column generation work?**
>
> The reason we can apply column generation is because the reduction approach is based on a linear program (LP) with suitable structural properties. It is important to note that these properties are *completely independent of the base classifier*, so the base classifier itself doesn't need to be based on linear programming and does not need to be amenable to column generation.
>
> There are two key properties of the LP in Eq. 10 that make it amenable to column generation:
> * Although the number of variables is intractably large, the number of constraints is small.
> * There is an efficient algorithm for finding the most violated constraint in the dual problem. (Primal variables correspond to dual constraints, and finding the most violated dual constraint can be achieved by calling the base classifier on the reweighted data, as is done in the $\\text{Best}_h$ function.)
>
> We will update our manuscript with this explanation.
>
> **(1C) Section 4.2: Is the setting of $Y_i$ in $\\text{Best}_h$ valid?**
>
> The setting $Y_i=1\\{w_i>0\\}$ follows exactly the reduction from cost-sensitive classification to weighted classification, as presented in Agarwal et al. 2018, Section 3.1. Specifically, note that our signed weights $w_i$ correspond to $C_i^0-C_i^1$ in their notation, and so the cost-sensitive problem that comes up in $\\text{Best}_h$ reduces to the weighted classification problem with weights $|w_i|=|C_i^0-C_i^1|$ and labels $1\\{w_i>0\\}=1\\{C_i^0>C_i^1\\}$.
>
> We will add a more detailed derivation to the manuscript (also as part of addressing the point 1F below).
>
> **(1D) Section 5.1: Explore trade-offs in ZAFAR DI/EO**
>
> The reviewer is right that regularization / penalty based approaches (including ZAFAR DI/EO) have a natural parameter to tune that would allow exploration of the accuracy/fairness frontier. (Similarly, post-processing can be adapted to explore different tradeoffs, as in [Cruz-Hardt, 2024](https://openreview.net/forum?id=jr03SfWsBS).) We will mention this in the paper.
>
> However, note that the focus of our paper is the speedup of ExpGrad. Our reason for including ZAFAR DI/EO and other baselines was not because we aim to argue that ExpGrad is better than these in terms of fairness or accuracy; such comparisons were focus of the original papers about reduction-based approaches to algorithmic fairness, and as far as we know none of the more recent research has invalidated the basic finding, supported by theory, which states that reduction-based approaches provably achieve optimal fairness-accuracy trade-offs. We included various other methods as baselines to help contextualize various performance metrics of ExpGrad and ExpGrad++. We believe that this purpose of contextualizing the performance metrics is served without a more detailed exploration of tradeoffs in those techniques. We are happy to point this out in our "Experimental evaluation" section. (Please also see our general comments regarding adding more experiments.)
>
> **(1E) Title change suggestion**
>
> The point is well taken. We wanted to be concise, but perhaps we went too far. We'll consider this.
>
> **(1F) Section 3.2: add explicit expressions for $\\mathbf{A}$, $\\mathbf{b}$, $\\mathbf{c}$, $c_0$**
>
> Will do.

---

> > ### Comment · Reviewer_9xny · 2026-04-03
> >
> > Thank you for responses. I felt that most of responses are reasonable.
> >
> > Additional questions and comments on the responses:
> >
> > - (1A): What are the difficulty to extend this method to deterministic classifiers?
> > - (1D): The issue that the description in the paper "ZAFAR DI/EO is only applicable to logistic regression" is clearly wrong for ZAFAR DI is also the point. At least this description should be fixed.
> > - (1D): To emphasize the paper's contribution for speeding up, how to show the results on fairness and speed is crucial. For example, (I am not sure but) if ExpGrad++ is faster than other methods (like ZAFAR), then it is good to show that the fairness values are better than or comparable to them. Or if ExpGrad++ is faster than ExpGrad but not faster than other methods, it is good to show that the fairness values are better than them in many of the cases. Such discussions are desired.

---

> > > ### Author Response · Authors · 2026-04-06
> > > **follow-up**
> > >
> > > Two follow-ups:
> > >
> > > (1A) Let me try to clarify. The reduction approach works with any base classifier (regardless of whether the classifier is randomized or deterministic), so reduction approach is applicable to both deterministic and randomized classifiers. However, reduction approach always returns a randomized classifier (even if the base classifier is deterministic).
> > >
> > > Column generation is a general approach for speeding up solving of linear programs. It can be used to speed up any classification algorithm that can be phrased as a linear program (regardless of whether the classification algorithm is randomized or deterministic). Reduction approach is an example of an algorithm that is written as a linear program; in this case, column generation yields a solution that is interpreted as a randomized classifier. On the other hand, [LPBoost](https://en.wikipedia.org/wiki/LPBoost) is also an example of a classification algorithm that can be written as a linear program; in this case, the solution of the linear program is interpreted as a deterministic classifier (and column generation can be applied to LPBoost). So column generation can be used to train both deterministic and randomized classifiers.
> > >
> > > That said, focusing on the reduction approach, it is not clear how to adjust the underlying linear program to yield a deterministic classifier (for example, simply interpreting the resulting ensemble as a deterministic classifier can yield a classifier that violates the fairness bound even though the originally returned randomized classifier satisfies it).
> > >
> > > Does this make sense?
> > >
> > > (1D) Thanks for pointing out incorrectness of "only applicable to logistic regression" (and apologies for not responding to it in the earlier reply). Our incorrect statement was most likely in reference to the specific library implementation that we used. We will fix it in the final version.

---

> > > > ### Comment · Reviewer_9xny · 2026-04-07
> > > >
> > > > Thanks, it means that this paper is based on a method developed for non-deterministic classifiers (reduction-based approach), so it may be alright that the paper itself does not consider extension to deterministic classifiers.

---

> > > > > ### Author Response · Authors · 2026-04-10
> > > > > **updates related to your requests**
> > > > >
> > > > > To confirm the last point: Yes, the ExpGrad method returns randomized classifiers, and our paper just seeks to speed up rather than extend ExpGrad.
> > > > >
> > > > > We have just uploaded a revision that we hope addresses some of our concerns:
> > > > > * We clarified that we have used implementation of Zafar DI/EO that is limited to logistic regression (hence that's what we evaluate).
> > > > > * We added additional tradeoff points for Zafar DI (see Figure 4a in Appendix A).

---

### Decision · Action_Editor_BqUG · 2026-04-26

**Recommendation:** Accept as is

**Additional Comments:**

There were some concerns with the original manuscript, including comparison with only older techniques and clarifications on the sampling approach. I appreciate the authors' efforts to address these concerns through the two rounds of revisions and manuscript updates.

All the reviewers recommend accepting the paper after the revisions, and no further changes are necessary for the camera-ready (please make sure to change the revision text color to black).

**Audience:**

Yes

**Audience Explanation:**

Given that reduction-based in-processing approaches are a widely used class for imposing statistical fairness, the paper's effort to speed up such approaches would be of interest to the TMLR community that focuses on the societal aspects of machine learning systems.

**Claims And Evidence:**

Yes

**Claims Explanation:**

The paper studies the popular class of in-processing approaches that satisfy statistical notions of fairness using a reduction-based approach. The authors show how to speed up such approaches without sacrificing the quality of the solution significantly. The claims are clearly mentioned, and sufficient evidence is provided.